# Extrinsic activin signaling cooperates with an intrinsic temporal program to increase mushroom body neuronal diversity

**Anthony M Rossi\*, Claude Desplan\***

Department of Biology, New York University, New York, United States

**Abstract** Temporal patterning of neural progenitors leads to the sequential production of diverse neurons. To understand how extrinsic cues influence intrinsic temporal programs, we studied *Drosophila* mushroom body progenitors (neuroblasts) that sequentially produce only three neuronal types: γ, then α′β′, followed by αβ. Opposing gradients of two RNA-binding proteins Imp and Syp comprise the intrinsic temporal program. Extrinsic activin signaling regulates the production of α′β′ neurons but whether it affects the intrinsic temporal program was not known. We show that the activin ligand Myoglianin from glia regulates the temporal factor Imp in mushroom body neuroblasts. Neuroblasts missing the activin receptor Baboon have a delayed intrinsic program as Imp is higher than normal during the α′β′ temporal window, causing the loss of α′β′ neurons, a decrease in αβ neurons, and a likely increase in γ neurons, without affecting the overall number of neurons produced. Our results illustrate that an extrinsic cue modifies an intrinsic temporal program to increase neuronal diversity.

## Introduction

The building of intricate neural networks during development is controlled by highly coordinated patterning programs that regulate the generation of different neuronal types in the correct number, place and time. The sequential production of different neuronal types from individual progenitors, *i. e.* temporal patterning, is a conserved feature of neurogenesis (*Cepko, 2014*; *Holguera and Desplan, 2018*; *Kohwi and Doe, 2013*; *Lodato and Arlotta, 2015*). For instance, individual radial glia progenitors in the vertebrate cortex sequentially give rise to neurons that occupy the different cortical layers in an inside-out manner (*Gao et al., 2014*; *Llorca et al., 2019*). In *Drosophila*, neural progenitors (called neuroblasts) also give rise to different neuronal types sequentially. For example, projection neurons in the antennal lobe are born in a stereotyped temporal order and innervate specific glomeruli (*Jefferis et al., 2001*; *Kao et al., 2012*; *Yu et al., 2010*). In both of these examples, individual progenitors age concomitantly with the developing animal (e.g., from embryonic stages 11–17 in mouse and from the first larval stage (L1) to the end of the final larva stage (L3) in *Drosophila*). Thus, these progenitors are exposed to changing environments that could alter their neuronal output. Indeed, classic heterochronic transplantation experiments demonstrated that young cortical progenitors placed in an old host environment alter their output to match the host environment and produce upper-layer neurons (*Desai and McConnell, 2000*; *McConnell, 1988*; *McConnell and Kaznowski, 1991*).

The adult *Drosophila* central brain is built from ~100 neuroblasts (*Lee et al., 2020*; *Urbach and Technau, 2004*; *Wong et al., 2013*; *Yu et al., 2013a*) that divide continuously from L1 to L3 (*Homem et al., 2014*; *Sousa-Nunes et al., 2010*; *Yang et al., 2017*). Each asymmetric division regenerates the neuroblast and produces an intermediate progenitor called ganglion mother cell (GMC) that divides only once, typically producing two different cell types (*Lin et al., 2010*; *Spana and Doe, 1996*; *Truman et al., 2010*). Thus, during larval life central brain

**\*For correspondence:**
amr808@nyu.edu (AMR);
cd38@nyu.edu (CD)

neuroblasts divide 50–60 times, sequentially producing many different neuronal types. All central brain neuroblasts progress through opposing temporal gradients of two RNA-binding proteins as they age: IGF-II mRNA binding protein (Imp) when they are young and Syncrip (Syp) when they are old (*Liu et al., 2015*; *Syed et al., 2017b*; *Syed et al., 2017a*; *Yang et al., 2016*). Loss of Imp or Syp in antennal lobe or Type II neuroblasts affects the ratio of young to old neuronal types (*Liu et al., 2015*; *Ren et al., 2017*). Imp and Syp also affect neuroblast lifespan (*Yang et al., 2017*). Thus, a single temporal program can affect both the diversity of neuronal types produced and their numbers.

Since central brain neuroblasts produce different neuronal types through developmental time, roles for extrinsic cues have recently garnered attention. Ecdysone triggers all the major developmental transitions including progression into the different larval stages and entry in pupation (*Yamanaka et al., 2013*). The majority of central brain neuroblasts are not responsive to ecdysone until mid-larval life when they begin to express the Ecdysone Receptor (EcR) (*Syed et al., 2017a*). Expressing a dominant-negative version of EcR (EcR-DN) in Type II neuroblasts delays the Imp to Syp transition that normally occurs ~60 hr after larval hatching (ALH). This leads to many more cells that express the early-born marker gene Repo and fewer cells that express the late-born marker gene Bsh.

To further understand how extrinsic signals contribute to temporal patterning, we studied *Drosophila* mushroom body neuroblasts because of the deep understanding of their development. The mushroom body is comprised of ~2000 neurons (Kenyon cells) that belong to only three main neuronal types that have unique morphologies and play distinct roles in learning and memory (*Cognigni et al., 2018*; *Ito et al., 1997*; *Lee et al., 1999*). They receive input mainly from ~200 projection neurons that each relays odor information from olfactory receptor neurons (*Vosshall and Stocker, 2007*). Each projection neuron connects to a random subset of Kenyon cells and each Kenyon cell receives input from ~7 different projection neurons (*Jefferis et al., 2007*; *Murthy et al., 2008*; *Turner et al., 2008*). This connectivity pattern requires a large number of mushroom body neurons (~2,000) to represent complex odors (*Hige, 2018*). To produce this very large number of neurons, mushroom body development is unique in many respects. Mushroom body neurons are born from four identical neuroblasts that divide continuously (unlike any other neuroblast) from the late embryonic stages until the end of pupation (~9 days for ~250 divisions each) (*Figure 1A*; *Ito et al., 1997*; *Kraft et al., 2016*; *Kunz et al., 2012*; *Kurusu et al., 2009*; *Lee et al., 1999*; *Pahl et al., 2019*; *Siegrist et al., 2010*; *Sipe and Siegrist, 2017*). Furthermore, the two neurons born from each mushroom body GMC are identical. The neuronal simplicity of the adult mushroom body makes it ideal to study how extrinsic cues might affect diversity since the loss of any single neuronal type is obvious given that each is represented hundreds of times.

The three main neuronal types that make up the adult mushroom body are produced sequentially during neurogenesis: first γ, followed by α′β′, and then αβ neurons (*Lee et al., 1999*; *Figure 1A*), representing the simplest lineage in the central brain. The γ temporal window extends from L1 (the first larval stage) until mid-L3 (the final larval stage) when animals attain critical weight and are committed to metamorphosis; the α′β′ window from mid-L3 to the beginning of pupation, and the αβ window from pupation until eclosion (the end of development). Like all other central brain neuroblasts Imp and Syp are expressed by mushroom body neuroblasts, but in much shallower gradients through time, which accounts for their extended lifespan (*Liu et al., 2015*; *Yang et al., 2017*). Imp and Syp are inherited by newborn neurons where they instruct temporal identity. Imp positively and Syp negatively regulate the translation of *chronologically inappropriate morphogenesis* (*chinmo*), a gene encoding a transcription factor that acts as a temporal morphogen in neurons (*Kao et al., 2012*; *Ren et al., 2017*; *Zhu et al., 2006*). The first-born γ neurons are produced for the first ~85 cell divisions, when Imp levels in neuroblasts, and thus Chinmo in neurons, are high. α′β′ neurons are produced for the next ~40 divisions, when Imp and Syp are at similar low levels that translate into lower Chinmo levels in neurons. Low Chinmo then regulates the expression in neurons of *maternal gene required for meiosis* (*mamo*), which encodes a transcription factor that specifies the α′β′ fate and whose mRNA is stabilized by Syp (*Liu et al., 2019*). αβ neurons are generated for the final ~125 neuroblast divisions, when Syp levels are high, Imp is absent in neuroblasts, and thus Chinmo and Mamo are no longer expressed in neurons.

Extrinsic cues are known to have important roles in regulating neuronal differentiation during mushroom body neurogenesis. The ecdysone peak that controls entry into pupation regulates γ

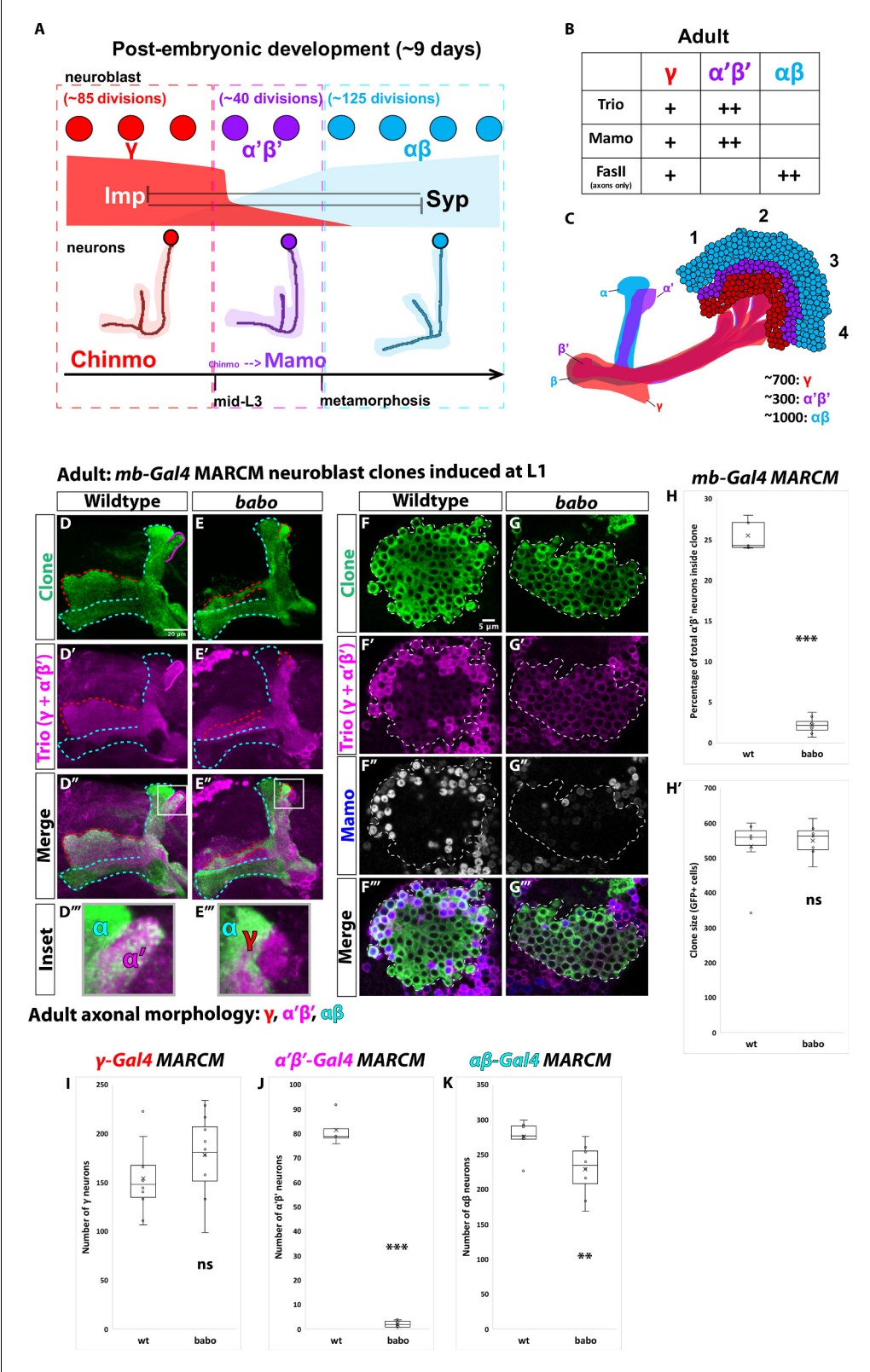

**Figure 1.** α'β' neurons are not generated from *babo* mutant neuroblasts. (**A**) Summary of intrinsic temporal patterning mechanism operating during mushroom body development. During early larval stages, mushroom body neuroblasts express high levels of Imp (red) and Chinmo (red) in neurons to specify γ identity for ~85 neuroblast divisions (red-dashed box). From mid-L3 to metamorphosis, when Imp and Syp (cyan) are both at low levels, the same neuroblast divides ~40 times to produce α'β' neurons (magenta-dashed box). Low Chinmo regulates the expression Mamo, a terminal selector of

*Figure 1 continued on next page*

Figure 1 continued

α'β' identity. From the beginning of metamorphosis throughout pupal development, high Syp leads to αβ neurons (cyan-dashed outline). (B) Known molecular markers can distinguish between the three mushroom body neuronal types in the adult. (C) Mushroom body projections originating from neurons born from four neuroblasts (numbered 1 to 4) per hemisphere fasciculate into a single bundle (peduncle) before branching into the five mushroom body lobes. The first-born γ neurons (red) remodel during development to project into a single, medial lobe in the adult. This lobe is the most anterior of the medial lobes. Axons from α'β' neurons (magenta) bifurcate to project into the vertical and medial α' and β' lobes. The β' lobe is posterior to the γ lobe. The last-born αβ neurons (cyan) also bifurcate their axons into the vertical projecting α lobe and medial projecting β lobe. The α lobe is positioned adjacent and medial to the α' lobe. The β lobe is the most posterior medial lobe. (D-E) Representative max projections showing adult axons of clonally related neurons born from L1 stage in wildtype and *babo* conditions. *UAS-CD8::GFP* is driven by *mb-Gal4* (*OK107-Gal4*). Outlines mark GFP$^+$ axons, where γ axons are outlined in red, α'β' axons are outlined in magenta, and αβ axons are outlined in cyan. A white box outlines the Inset panel. Trio (magenta) is used to label all γ and α'β' axons for comparison to GFP$^+$ axons. (D) In wildtype, GFP$^+$ axons (green, outlined in red, magenta and cyan) are visible in all observable mushroom body lobes. (E) In *babo* mutant clones, γ neurons (red outline) remain unpruned. GFP$^+$ axons are missing inside the Trio$^+$ α' lobe, indicating the absence of α'β' neurons. (F-G) Representative, single z-slices from the adult cell body region of clones induced at L1 in wildtype and *babo* conditions. *UAS-CD8::GFP* is driven by *mb-Gal4*. (F) Wildtype clones show the presence of strongly expressing Trio (magenta) and Mamo (blue, gray in single channel) neurons, indicative of α'β' identity. (G) In *babo* mutant clones, cells strongly expressing Trio and Mamo are not present. (H) Quantification of MARCM clones marked by *mb-Gal4*, which labels all mushroom body neuronal types. The number of α'β' neurons are quantified in wildtype (n = 7) and *babo* (n = 8) conditions. Plotted is the percentage of strong Mamo$^+$ and GFP$^+$ cells (clonal cells) versus all Mamo$^+$ cells (clonal and non-clonal cells) within a single mushroom body. In wildtype, 25.5 ± 0.7% of the total strong Mamo expressing cells (α'β' neurons) are within clones, consistent with our expectation since each mushroom body is made from four neuroblasts. In *babo* clones, only 2.2 ± 0.4% of α'β' neurons are within clones. (H') There are no significant differences between the average clone sizes (wildtype:533.6 ± 33.3; *babo*:551.3 ± 17.6). (I) Quantification of γ neurons marked by γ-*Gal4* (*R71G10-Gal4*) in MARCM clones. Plotted is the total number of γ neurons marked by GFP and Trio in wildtype (n = 10) and *babo* mutant (n = 12) clones. In wildtype, the average number of γ neurons is 154.3 ± 11.4. In *babo* mutants, the average is 178.4 ± 11.9. (J) Quantification of α'β' neurons marked by α'β'-*Gal4* (*R41C07-Gal4*) in MARCM clones. Plotted is the total number of α'β' neurons marked by GFP and strong Trio in wildtype (n = 4) and *babo* mutant (n = 8) clones. In wildtype, the average number of α'β' neurons is 81.5 ± 3.4. In *babo* mutants, the average is 2.1 ± 0.5. (K) Quantification of αβ neurons marked by αβ-*Gal4* (*R44E04-Gal4*) in MARCM clones. Plotted is the total number of GFP$^+$ cells in wildtype (n = 7) and *babo* mutant (n = 8) clones. In wildtype, the average number is 276 ± 9.1. In *babo* mutants, the average number is 228.9 ± 13.2. A two-sample, two-tailed t-test was performed. ***p<0.001, **p<0.01, ns: not significant. Scale bars: D, 20 μm; F, 5 μm.

The online version of this article includes the following source data and figure supplement(s) for figure 1:

**Source data 1.** Neuron number counts for data presented in *Figure 1A* and *Figure 1—figure supplement 1*.
**Figure supplement 1.** α'β' neurons are lost from the adult neuropil in activin signaling mutant clones.
**Figure supplement 2.** γ neuron numbers likely increase, while αβ numbers decrease, in *babo* mutant clones.

neuron axonal remodeling (*Lee et al., 2000*). Ecdysone was also proposed to be required for the final differentiation of α'β' neurons (*Marchetti and Tavosanis, 2017*). EcR expression in γ neurons is timed by activin signaling, a member of the TGFβ family, from local glia (*Awasaki et al., 2011*; *Zheng et al., 2003*). Activin signaling from glia is also required for the α'β' fate (*Marchetti and Tavosanis, 2019*): Knocking-down the activin pathway receptor Baboon (Babo) leads to the loss of α'β' neurons. It was proposed that activin signaling in mushroom body neuroblasts regulates the expression of EcR in prospective α'β' neurons and that when the activin pathway is inhibited, it leads to the transformation of α'β' neurons into later-born pioneer-αβ neurons (a subclass of the αβ class) (*Marchetti and Tavosanis, 2019*).

Although there is strong evidence that extrinsic cues have important functions in neuronal patterning in the *Drosophila* central brain, it remains unknown how extrinsic temporal cues interface with the Imp and Syp intrinsic temporal program to regulate neuronal specification. Here we address this question using the developing mushroom bodies. We independently discovered that activin signaling from glia is required for α'β' specification. However, we show that activin signaling lowers the levels of the intrinsic factor Imp in mushroom body neuroblasts to define the mid-α'β' temporal identity window. Removing the activin receptor Babo in mutant clones leads to the loss of α'β' neurons, to fewer last-born αβ neurons, and to the likely generation of additional first-born γ neurons without affecting overall clone size. This appears to be caused by a delayed decrease in Imp levels, although the intrinsic temporal clock still progresses even in the absence of activin signaling. We also demonstrate that ecdysone signaling is not necessary for the specification of α'β' neurons, although it might still be involved in later α'β' differentiation. Our results provide a model for how intrinsic and extrinsic temporal programs operate within individual progenitors to regulate neuronal specification.

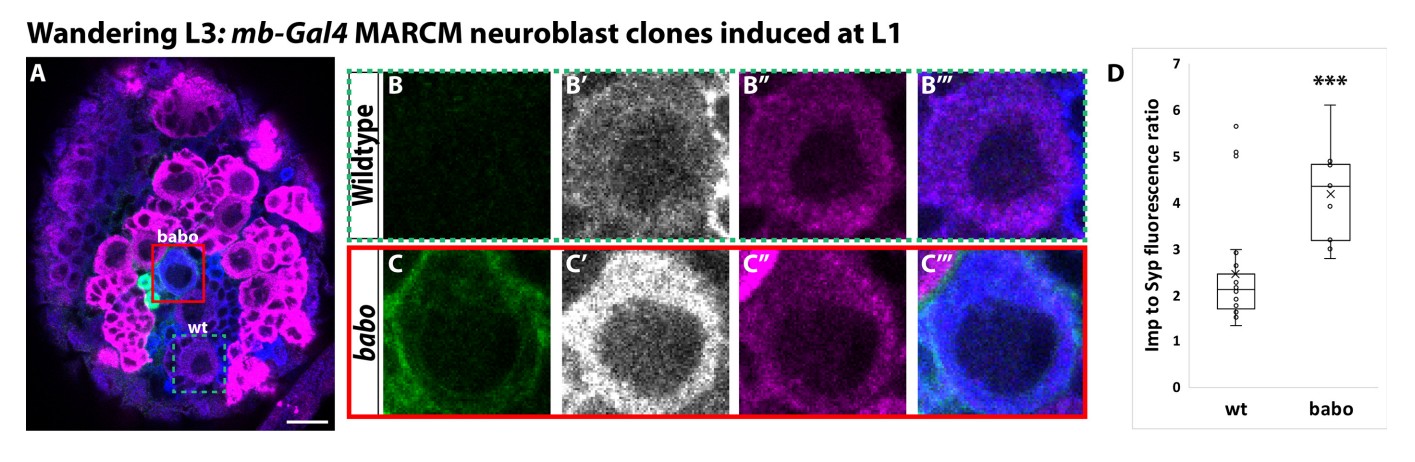

**Figure 2.** Activin signaling acts in neuroblasts to lower Imp levels. (**A**) Representative image of a *babo* mushroom body neuroblast marked by *UAS-CD8::GFP* driven by *mb-Gal4* (red box) adjacent to a wildtype neuroblast (green-dashed box) in the same focal plane from a wandering L3 stage brain, immunostained for Imp (blue, gray in single channel) and Syp (magenta). (**B**) Close-up view of wildtype neuroblast (green-dashed box in A). (**C**) Close up view of *babo* mutant neuroblast (red box in A). (**D**) Quantification of the Imp to Syp ratio in *babo* neuroblasts (4.2 ± 0.4, n = 9 from 4 different brains) compared to wildtype (2.4 ± 0.3, n = 23 from the same 4 brains as *babo* neuroblasts). A two-sample, two-tailed t-test was performed. ***p<0.001, ns: not significant. Scale bar: 10 μm.

The online version of this article includes the following source data and figure supplement(s) for figure 2:

**Source data 1.** Imp and Syp fluorescence quantification in *babo* mutant clones.
**Figure supplement 1.** Activin signaling lowers Imp levels but the Imp to Syp transition does not depend on activin signaling.

# Results

## α′β′ neurons are not generated from *babo* mutant neuroblasts

The production of the three different mushroom body neuronal types occurs within specific developmental stages of larval and pupal development. That is, the γ window extends from L1 to mid-L3, the α′β′ window from mid-L3 to pupation, and the αβ window from pupation to eclosion (*Figure 1A*; *Lee et al., 1999*). This means that extrinsic cues could play a role in controlling or fine-tuning transitions between these temporal windows. Additionally, the specification of neuronal identity within each temporal window could be aided by extrinsic cues. To test these hypotheses, we used Mosaic Analysis with a Repressible Cell Marker (MARCM) (*Lee and Luo, 1999*) to test the function of receptors for inter-cellular signaling pathways with known roles either in mushroom body neurogenesis (Activin and Ecdysone) (*Lee et al., 2000*; *Marchetti and Tavosanis, 2017*; *Marchetti and Tavosanis, 2019*; *Zheng et al., 2003*) or more broadly during nervous system development (Hedgehog and juvenile hormone) (*Figure 1—figure supplement 1A–H*; *Baumann et al., 2017*; *Chai et al., 2013*). We induced mushroom body neuroblast clones at L1 and compared the axonal morphologies of adult neurons born from mutant neuroblasts to neurons born from surrounding wildtype neuroblasts. To identify mushroom body axonal lobes (both mutant and wildtype), we used antibodies to the Rho guanine exchange factor Trio (a weak γ and strong α′β′ cytoplasmic marker) and to the cell adhesion molecule Fasciclin II (FasII) (an axonal γ and αβ marker) (*Figure 1B*; *Awasaki et al., 2000*; *Crittenden et al., 1998*). To visualize mushroom body neurons within clones we expressed *UAS-CD8::GFP* under the control of *OK107-Gal4* (referred to as *mb-Gal4* hereafter), a Gal4 enhancer trap in *eyeless* and a common mushroom body Gal4 driver that strongly labels all mushroom body neuronal types during development and in the adult, and weakly mushroom body neuroblasts and young neurons throughout development (*Connolly et al., 1996*; *Liu et al., 2015*; *Zhu et al., 2006*).

In wildtype clones induced at L1, GFP⁺ axons projected to all five mushroom body lobes: α, α′, β, β′ (hidden behind the γ lobe in max projections), and γ (*Figure 1D*, *Figure 1—figure supplement 1A*). In clones mutant for *babo,* we did not detect GFP⁺ axons within the α′β′ lobes, which remained visible by Trio staining due to the presence of wildtype α′β′ neurons (*Figure 1E*, *Figure 1—figure supplement 1B*). In addition, and as previously described, γ neurons within *babo* mutant clones

remained unpruned (visualized by vertical GFP⁺ axons that were Trio⁺ and FasII⁺), providing a positive control since γ remodeling is known to require activin signaling (**Figure 1E**, **Figure 1—figure supplement 1B**; **Awasaki et al., 2011**; **Yu et al., 2013b**; **Zheng et al., 2003**).

Babo is the sole Type I receptor in the activin pathway (a member of the TGFβ family of signaling molecules). Babo with its Type II co-receptors binds four different activin ligands and acts through the transcription factor Smad on X (Smad2) (**Brummel et al., 1999**; **Upadhyay et al., 2017**). We induced *Smad2* mutant clones at L1 and characterized adult axonal morphologies. Similar to *babo* mutant clones, *Smad2* clones were missing α′β′ neurons and also contained unpruned γ neurons (**Figure 1—figure supplement 1I**).

The absence of GFP⁺ axons within the α′β′ lobes in *babo* mutant clones could be due to the loss of axonal projections, or to the loss of neuronal identity. Using antibodies against Trio and Mamo that strongly label α′β′ neuron cell bodies in the adult (**Figure 1B**, **Figure 1—figure supplement 1J**; **Alyagor et al., 2018**; **Awasaki et al., 2000**; **Croset et al., 2018**; **Liu et al., 2019**), we detected strong Trio⁺ and Mamo⁺ cells within adult GFP⁺ clones induced at L1 (**Figure 1F**, **Figure 1—figure supplement 1K**). In *babo* mutant clones however, the vast majority of strong Trio⁺ and Mamo⁺ cells inside clones were missing compared to surrounding wildtype neurons (**Figure 1G**, **Figure 1—figure supplement 1L**), suggesting that α′β′ neurons were not specified. We quantified the number of α′β′ neurons in wildtype and *babo* clones by counting the number of strong Mamo⁺ cells within a clone versus the total number of strong Mamo⁺ cells outside the clone. In wildtype MARCM clones affecting a single mushroom body neuroblast (n = 7), the percentage of all α′β′ neurons that were present within the clones was 25.5 ± 0.7%, the expected ratio since each mushroom body is built from four identical neuroblasts (**Figure 1H**; **Ito et al., 1997**; **Lee et al., 1999**). In comparison, in *babo* mutant clones (n = 8) the percentage of α′β′ neurons within clones was 2.2 ± 0.4% (**Figure 1H**). Interestingly, although there was a decrease in the number of adult α′β′ neurons upon expression of *UAS-babo-RNAi* with *mb-Gal4*, the majority of α′β′ neurons were not lost (**Figure 1—figure supplement 1M–Q**). Importantly, γ neurons in these brains did not remodel (**Figure 1—figure supplement 1P–P′′′**), indicating that the *babo-RNAi* worked efficiently. This difference with *babo* clones is likely due to the weak expression of *mb-Gal4* in neuroblasts and newborn neurons and suggested to us that activin signaling is necessary for α′β′ specification by acting in neuroblasts (see below).

We next sought to determine the fate of the missing α′β′ neurons in *babo* clones, particularly since there was no significant difference in average clone sizes between mutant and control clones labeled with *mb-Gal4* (wildtype: clone size = 533.6 ± 33.3, n = 7; *babo*: clone size = 551.3 ± 17.6, n = 7) (**Figure 1H′**), which suggests that there is no defect in neuroblast proliferation, and that α′β′ neurons are not lost by cell death in *babo* clones. However, to directly test whether cell death played a role, we expressed the caspase inhibitor P35 in *babo* mutant clones (**Figure 1—figure supplement 2A**). However, α′β′ neurons were still missing in the adult (**Figure 1—figure supplement 2A**), indicating that α′β′ neurons are not generated and then die. We thus tested whether the γ or αβ temporal windows were extended in *babo* mutant clones. We made MARCM clones in which the γ, α′β′ or αβ neurons were specifically marked with different *Gal4* lines, and then quantified the total number of GFP⁺ neurons in wildtype versus *babo* mutant clones (**Figure 1I–K**, **Figure 1—figure supplement 2B–M**). Using *R71G10-Gal4* (**Issman-Zecharya and Schuldiner, 2014**) (referred to as γ-*Gal4*), the average number of γ neurons trended higher in *babo* mutant clones, although not significantly (wildtype: 154.3 ± 11.4, n = 10; *babo*: 178.4 ± 11.9, n = 12) (**Figure 1I**, **Figure 1—figure supplement 2B–E**), likely because the number of γ neurons directly depends on the time of clone induction. α′β′ neurons, marked by *R41C07-Gal4* (referred to as α′β′-*Gal4*), were mostly missing in *babo* mutant clones compared to wildtype clones (wildtype: 81.5 ± 3.6, n = 4; *babo*: 2.1 ± 0.5, n = 8,) consistent with our previous results when counting strong Mamo⁺ cells in *babo* clones marked by *mb-Gal4* (**Figure 1H**, **Figure 1J**, **Figure 1—figure supplement 2F–I**). The average number of αβ neurons, marked by *R44E04-Gal4* (referred to as αβ-*Gal4*), was significantly reduced in *babo* versus wildtype clones (wildtype: 276 ± 9.1, n = 7,; *babo*: 228.9 ± 13.2, n = 8) (**Figure 1K**, **Figure 1—figure supplement 2J–M**). Together, these results suggest that additional γ neurons are likely produced, and that fewer αβ neurons are generated, in *babo* mutant clones. Although this might appear to contradict a recent report that showed that a subclass of αβ neurons (pioneer-αβ) increases (**Marchetti and Tavosanis, 2019**), it is likely that the earliest born pioneer-αβ are increased because of the delayed decrease in the Imp gradient, but that the later-born αβ neurons do not have time to form (see below and Discussion). We note that the total number of neurons labeled by

our neuron type specific *Gal4* drivers did not add up to the expected number of ~500 neurons in *babo* mutant clones, which is likely explained by the large variability in the number of γ neurons labeled by γ-*Gal4*. Next, we focused on understanding whether and how activin signaling interacts with the intrinsic Imp and Syp temporal program.

## Activin signaling acts in neuroblasts to lower Imp levels and specify α′β′ neurons

Given that α′β′ neuronal specification is intrinsically controlled by Imp and Syp (*Liu et al., 2015*), we asked whether activin signaling acts through or in parallel to this intrinsic temporal system, specifically at L3 when α′β′ neurons are being produced. We first asked whether *babo* is expressed at L3 in mushroom body neuroblasts. Based on published transcriptome data collected from mushroom body neuroblasts at different developmental stages (*Liu et al., 2015*; *Yang et al., 2016*), *babo* is expressed evenly through time in mushroom body neuroblasts, unlike the two RNA binding proteins *Imp* and *Syp* (*Figure 2—figure supplement 1A*). Although this measure does not take into account the possibility of post-transcriptional regulation, it is likely that the activin signaling pathway is temporally controlled by ligand interaction and not by differential expression of *babo*.

To directly test whether activin signaling acts on Imp and Syp to affect α′β′ specification, we induced MARCM clones for *babo* at L1 and compared the Imp to Syp protein ratio in mutant mushroom body neuroblasts to surrounding wildtype neuroblasts at wandering L3 (*Figure 2*). The average Imp to Syp ratio was significantly higher in *babo* neuroblasts (ratio: 4.2 ± 0.4; n = 9 from 4 different brains) compared to wildtype neuroblasts (ratio: 2.4 ± 0.2; n = 23 from the same 4 brains as *babo*) at L3, driven by a significantly higher Imp level in mutant neuroblasts (*Figure 2*, *Figure 2—figure supplement 1B*) while Syp was not significantly different (*Figure 2—figure supplement 1B′*). In addition, the α′β′ neuronal marker Mamo (*Liu et al., 2019*) was lost in *babo* mutant clones at L3 while the level of Chinmo was higher in these neurons (*Figure 2—figure supplement 1C–C′′′*), consistent with the notion that high Imp levels block α′β′ specification through maintained higher levels of Chinmo that likely lead to the increased production of γ neurons. The significantly higher Imp to Syp ratio in *babo* mutant neuroblasts persisted even ~24 hr After Pupal Formation (APF) (*babo* ratio: 0.58 ± 0.11; n = 7 from 6 different brains; wildtype ratio: 0.27 ±. 02; n = 27 from the same 6 brains as *babo*), once again driven by higher Imp levels (*Figure 2—figure supplement 1D–I*). Together, these results indicate that activin signaling lowers Imp levels at late larval and early pupal stages. Importantly, although Imp was higher in *babo* mutant neuroblasts and persisted longer, the absolute level of Imp still decreased significantly albeit with prolonged kinetics, while the absolute level of Syp was higher in *babo* mutant neuroblasts at ~24 hr APF vs. L3 (*Figure 2—figure supplement 1J–J′*): Thus, these changes are either intrinsically regulated or are affected by additional extrinsic factors. Our finding that Imp and Chinmo were higher in *babo* mutant neuroblasts and neurons at L3 is also consistent with our suggestion that additional γ neurons are produced during the α′β′ time window. The lack of α′β′ neurons in *babo* mutant clones even though Imp levels were finally low at ~24 hr APF suggests that α′β′ specification may only occur from L3 to the start of pupation.

We have shown that activin signaling functions in mushroom body neuroblasts to decrease Imp during L3. However, previous studies have shown that Babo also acts post-mitotically in mushroom body γ neurons where it times the expression of EcR for their remodeling, indicating that Babo can act independently in neuroblasts and in neurons (*Zheng et al., 2003*). To test if activin signaling functions post-mitotically in prospective α′β′ neurons, we characterized the morphology of *babo* mutant neurons born from ganglion mother cell (GMC) clones induced during mid-late L3, the time at which α′β′ neurons are born. GMCs are intermediate progenitors that divide only once to produce two neurons. In this way, the role of Babo in prospective α′β′ neurons can be tested without affecting mushroom body neuroblasts: α′β′ neurons were present in *babo* GMC clones (n = 34/34), observable by axonal projections into the Trio labeled α′β′ lobes (*Figure 2—figure supplement 1K–K′*). As a positive control for the efficiency of *babo* GMC clones, we also made *babo* GMC clones at L1 to target γ neurons. In the majority of cases, γ axons remained unpruned (n = 8/10, *Figure 2—figure supplement 1L–L′*; *Zheng et al., 2003*). These results show that activin signaling acts in mushroom body neuroblasts, and not in neurons, to specify the α′β′ fate.

## Activin signaling is sufficient to expand production of α'β' neurons

Since activin signaling functions in mushroom body neuroblasts and is necessary for α'β' specification, we next investigated whether it is sufficient for the α'β' fate. We expressed a constitutively active form of the Babo receptor (*UAS-Babo-Act*) throughout development in MARCM clones with *mb-Gal4* and assessed the total number of α'β' neurons in the adult by strong Mamo expression. While in wildtype clones the percentage of α'β' neurons was 25.5 ± 0.7% (n = 7), the number of α'β' neurons present within *UAS-Babo-Act* clones significantly increased to 32 ± 1.4% (n = 4) (*Figure 3A–B*). To ask when these additional α'β' neurons were produced, we characterized the expression of the α'β' marker Mamo in young neurons at early L3, when γ neurons are being produced. In comparison to adjacent wildtype neurons, Mamo was expressed in neurons in *UAS-Babo-Act* clones at this stage (*Figure 3C–D*). These results confirm the precocious specification of α'β' neurons, likely at the expense of γ neurons (which would normally be produced at this stage). Importantly, constitutively expressing an activated version of Babo did not result in adult clones consisting entirely of α'β' neurons, highlighting that activin signaling alone is not master regulator of the α'β' fate and that other components are necessary to specify this neuronal type.

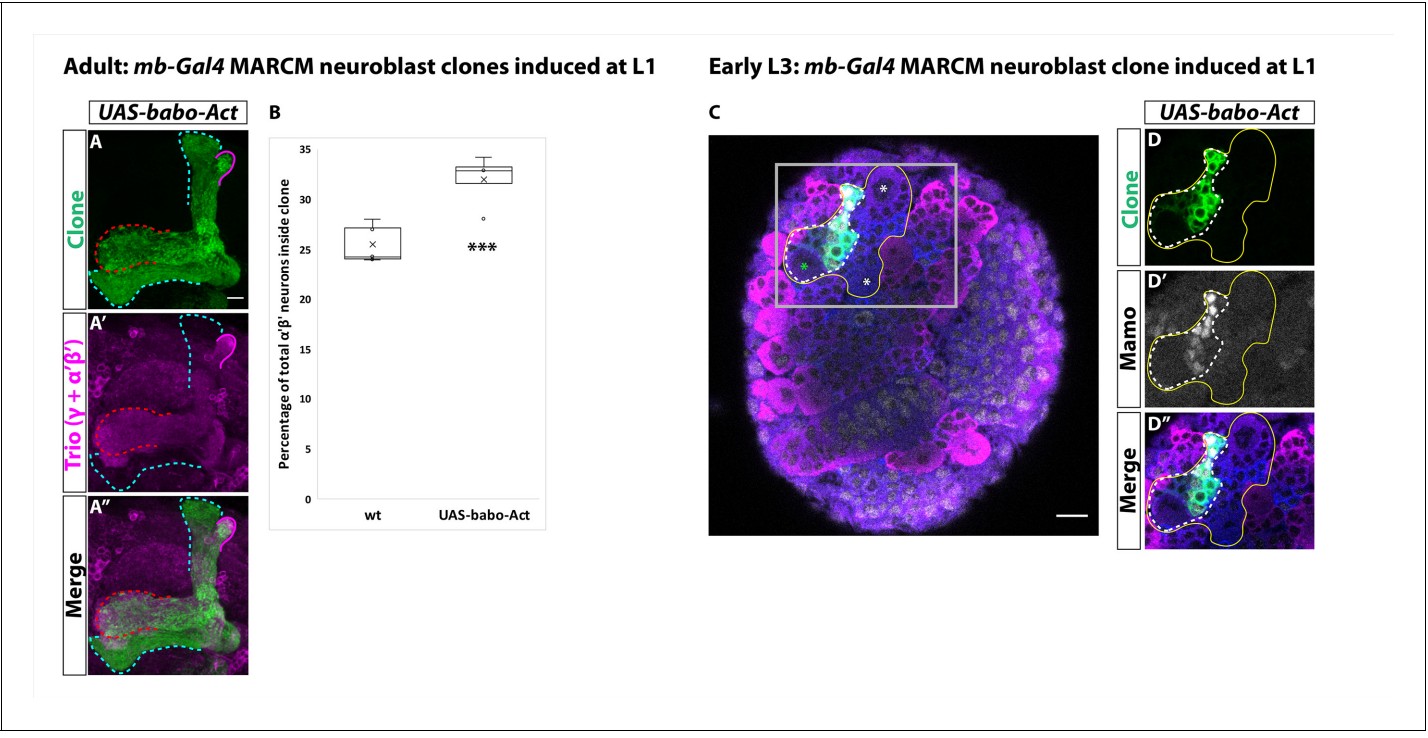

**Figure 3.** Activin signaling is sufficient to expand production of α'β' neurons. (**A**) Expression of *UAS-Babo-Act* by *mb-Gal4* leads to additional α'β' neurons but does not convert all mushroom body neurons into this fate. (**B**) Plotted is the percentage of strong Mamo+ and GFP+ cells (clonal cells) versus all Mamo+ cells (clonal and non-clonal cells) within a single mushroom body. The number of α'β' neurons is quantified in wildtype (n = 7, replotted from data in *Figure 1H*) and *UAS-babo-Act* (n = 4). In wildtype, 25.5 ± 0.7% of the total strong Mamo expressing cells (α'β' neurons) are within a clone while precociously activating the activin pathway increased the percentage to 32 ± 1.4%. (**C**) A representative image of an early L3 brain in which a single mushroom body neuroblast is expressing *UAS-babo-Act* driven by *mb-Gal4* (white-dashed line). Imp (blue) and Syp (magenta), along with GFP (green), are used to identify mushroom body neuroblasts (asterisks) and neurons. (**D**) Inset (gray box in C) showing that Mamo (gray) is expressed inside GFP+ cells that express *UAS-babo-Act* but not outside in adjacent wildtype mushroom body neurons (yellow line) (n = 3/3). A two-sample, two-tailed t-test was performed. ***p<0.001.

The online version of this article includes the following source data for figure 3:

**Source data 1.** Neuron number counts for data presented in *Figure 3*.

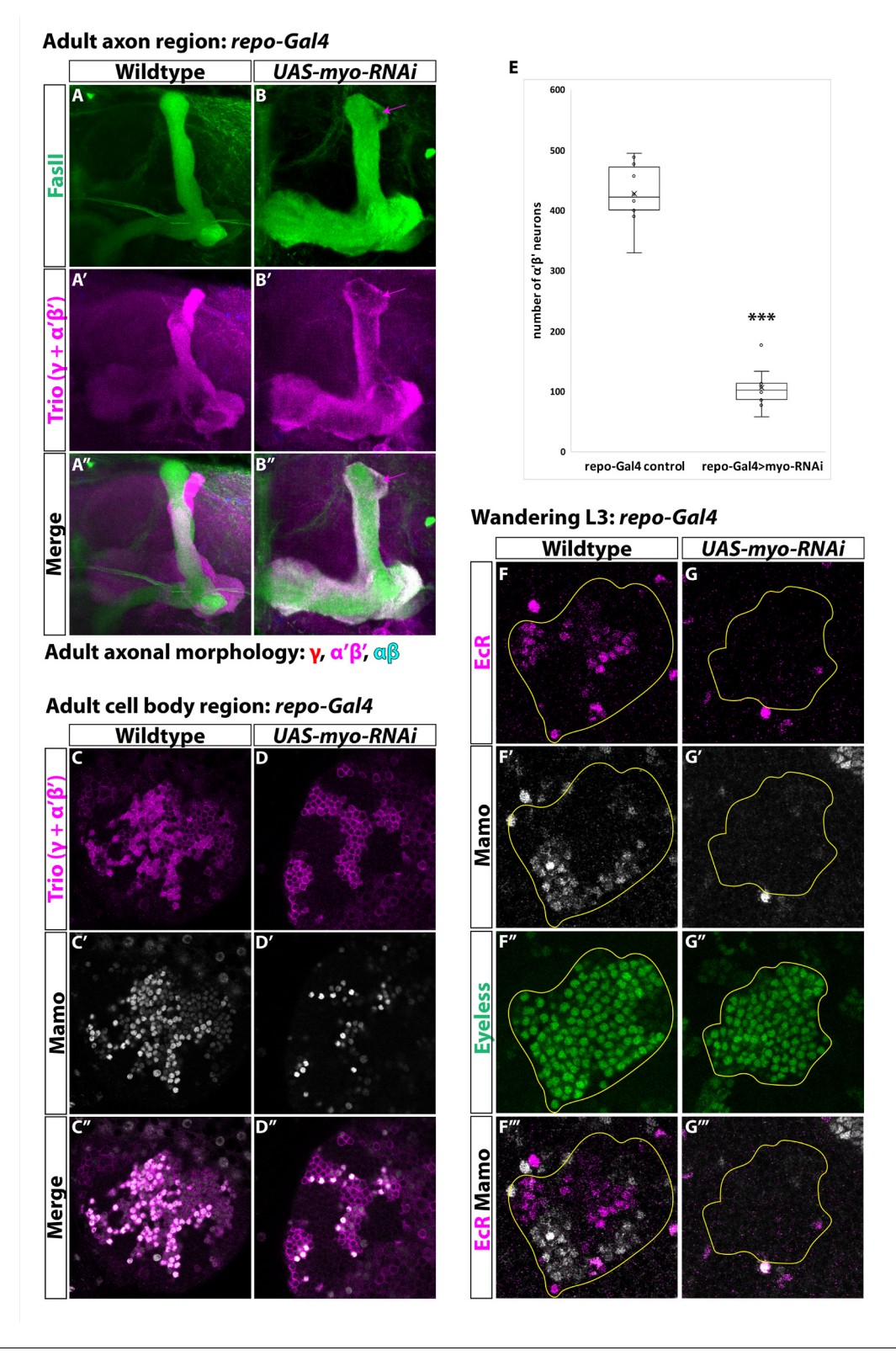

**Figure 4.** Glia are the source of the activin ligand Myo to specify α'β' neurons. (**A-B**) Representative images of adult mushroom body lobes labeled by FasII (green) and Trio (magenta). (**A**) In wildtype controls (428.9 ± 16.2, n = 10) (*repo-Gal4* only) all three neuronal types are present based on axonal projections. (**B**) Expressing *UAS-myo-RNAi* (106.6 ± 11.4, n = 10) causes γ neurons not to remodel and to the loss of the majority of α'β' neurons, however some still remain (purple arrow, FasII- region). (**C-D**) Representative images of adult mushroom body cell body region. Trio (magenta) and

*Figure 4 continued on next page*

*Figure 4 continued*

Mamo (gray) are used to distinguish between the three neuronal types. Expressing *UAS-myo-RNAi* leads to loss of the majority of strong Mamo+ and Trio[+] cells, indicating the loss of α'β' neurons. (E) Quantification of phenotypes presented in A-D. (F) At L3, EcR (magenta) and Mamo (gray) are expressed in mushroom body neurons labeled by Eyeless (green, yellow outline). Mamo[+] cells are newborn α'β' neurons. G. Expressing *UAS-myo-RNAi* with *repo-Gal4* leads to loss of both Mamo and EcR in mushroom body neurons. A two-sample, two-tailed t-test was performed. ***p<0.001.
The online version of this article includes the following source data for figure 4:

**Source data 1.** Neuron number counts for data presented in *Figure 4*.

## Glia are the source of the activin ligand Myoglianin to specify α'β' neurons

Our finding that activin signaling plays an important role in specifying α'β' identity during mushroom body development led us to question from where the activin ligand originates. Glia secrete the activin ligand Myoglianin (Myo) to initiate γ neuron remodeling by activating EcR at L3 (*Awasaki et al., 2011*; *Yu et al., 2013b*). Therefore, we hypothesized that Myo from glia may also regulate α'β' specification. To test this, we knocked-down *myo* by expressing *UAS-myo-RNAi* with *repo-Gal4*, a driver expressed in all glia, and quantified the total number of α'β' neurons based on strong Mamo expression in the adult (*Figure 4A–E*). In comparison to control (428.9 ± 16.2, n = 10), the number of α'β' neurons was dramatically reduced (106.6 ± 11.4; n = 10) (*Figure 4E*). Mamo was also not expressed in mushroom body neurons at L3 (*Figure 4F–G*). Importantly, EcR was not expressed in γ neurons at this stage, providing a positive control for the efficiency of *UAS-myo-RNAi* (*Awasaki et al., 2011*). We note that even though the number of α'β' neurons was reduced in this experiment, *myo* knockdown was weaker than *babo* mutant clones, possibly due to incomplete knockdown of *myo* or because more than one ligand (or more than one source) contribute to α'β' specification (see Discussion). Our results are consistent with a recent report that also showed that glia are the source of Myo for α'β' specification (*Marchetti and Tavosanis, 2019*).

## α'β' neurons are specified by low Imp levels at L3

We and others have shown that activin signaling is necessary for α'β' specification (*Marchetti and Tavosanis, 2019*). We have shown that activin signaling acts by lowering Imp levels at L3. Although Imp is required for α'β' specification (*Figure 5—figure supplement 1A–E*; *Liu et al., 2015*), we wanted to determine whether low Imp levels are required at L3. We therefore characterized Mamo expression in young neurons at L3 following knockdown (*UAS-Imp-RNAi*) or overexpression (*UAS-Imp-OE*) of Imp with *mb-Gal4*. (*Figure 5A–C*; *Liu et al., 2015*). Consistent with our model, Mamo was not expressed in either condition. In comparison, although knocking-down *Syp* by expressing *UAS-Syp-RNAi* led to the loss of Mamo, its early overexpression (*UAS-Syp-OE*) did not (*Figure 5D–E*). The loss of Mamo in *Syp* knockdown is consistent with its role in stabilizing *mamo* transcripts at L3 (*Liu et al., 2019*). We conclude that low Imp and low or high Syp levels are required for α'β' specification. Consistent with this, we were unable to rescue the loss of α'β' neurons in *babo* mutant clones by constitutively repressing Imp with *UAS-Imp-RNAi* (0.2 ± 0.2%, n = 7) (*Figure 5—figure supplement 1I*, *Figure 5—figure supplement 1N*, *Figure 5—figure supplement 1P*), likely due to Imp reduction below the threshold required for α'β' specification (see Discussion). However, we could rescue *babo* mutant clones by expressing *UAS-babo* (21.1 ± 2.4%, n = 6) (*Figure 5—figure supplement 1F–H*, *Figure 5—figure supplement 1K–M*, *Figure 5—figure supplement 1P*). Overexpressing Syp (*UAS-Syp-OE*) to reduce the altered Imp:Syp ratio in *babo* mutant clones also did not rescue α'β' neurons (1.8 ± 0.5%, n = 7) (*Figure 5—figure supplement 1J*, *Figure 5—figure supplement 1O–P*), further highlighting that Imp but not Syp levels are important for α'β' specification.

## Ecdysone signaling is not necessary for α'β' specification

It has been proposed that activin signaling in mushroom body neuroblasts leads to EcR expression in neurons and that ecdysone signaling at late L3 induces differentiation of α'β' neurons (*Marchetti and Tavosanis, 2017*; *Marchetti and Tavosanis, 2019*). The role of ecdysone was tested by expressing a dominant-negative ecdysone receptor (*UAS-EcR-DN*). We confirmed these results by also expressing *UAS-EcR-DN* driven by *mb-Gal4* and were unable to detect GFP[+] mutant axons within adult α'β' lobes marked by Trio (*Figure 6A–B*). In addition, strong Trio[+] and Mamo[+] cells

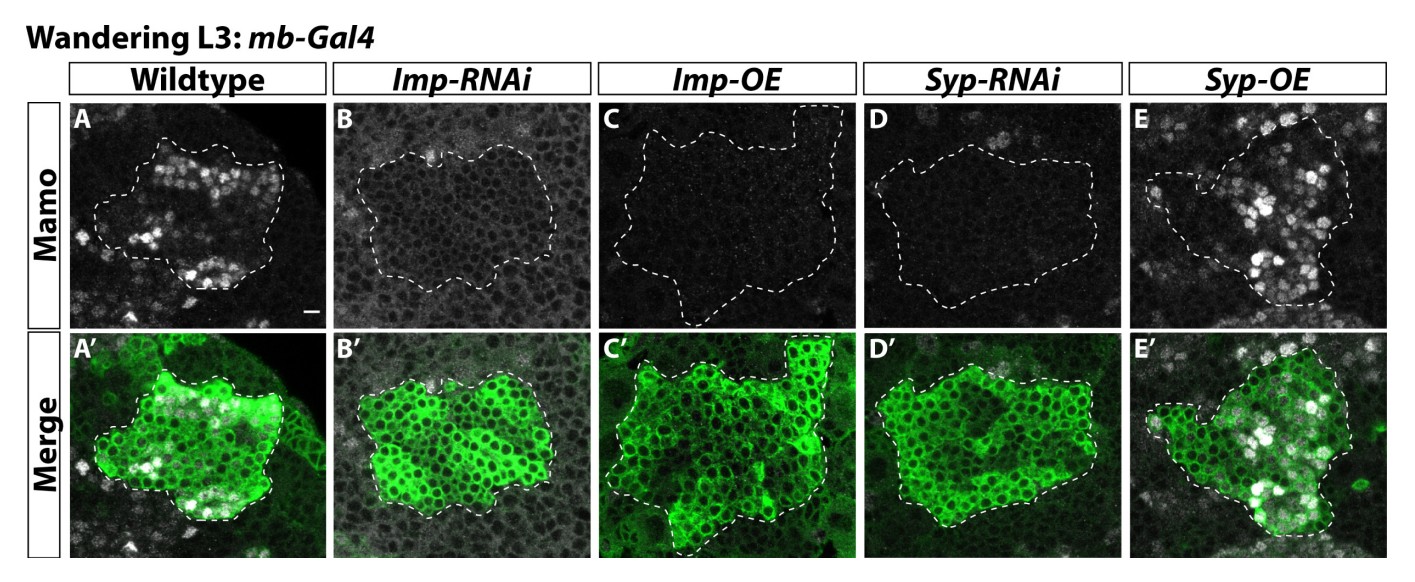

**Figure 5.** α'β' neurons are specified by low Imp levels at L3. (**A-A'**) Representative image of wildtype mushroom body neurons labeled by *mb-Gal4* driving *UAS-CD8::GFP* (green, white-dashed outline) during the wandering L3 stage. Mamo (gray) is used as a marker for α'β' neurons. (**B-B'**) When *mb-Gal4* is used to drive *UAS-Imp-RNAi*, Mamo is not expressed. (**C-C'**) Similarly, Mamo expression is lost when overexpressing *Imp* (*UAS-Imp-overexpression* (*OE*)). (**D-D'**). Expressing *UAS-Syp-RNAi* also leads to the loss of Mamo. (**E**) Expressing *UAS-Syp-overexpression* (*OE*) does not affect Mamo. Scale bar: 5 μm.

The online version of this article includes the following source data and figure supplement(s) for figure 5:

**Source data 1.** Neuron number counts for data presented in *Figure 5—figure supplement 1*.

**Figure supplement 1.** Low Imp levels are required for α'β' specification.

were missing inside *UAS-EcR-DN* mutant clones compared to wildtype clones (wildtype: 25.5 ± 0.7%, n = 7; *UAS-EcR-DN*: 3.4 ± 0.6%, n = 6) (*Figure 6C–E*, *Figure 6—figure supplement 1A–B*). However, we were surprised to find that α'β' neurons were still present in mutant clones for the EcR co-receptor *ultraspiracle* (*usp*) (*Figure 6F*, *Figure 6—figure supplement 1C*). Therefore, we sought to better understand how expressing *UAS-EcR-DN* blocks α'β' specification.

First, unlike our result in *babo* mutant neuroblasts, we did not observe a significant difference in the average Imp to Syp protein ratio at L3 in *UAS-EcR-DN* expressing mushroom body neuroblasts with *mb-Gal4* (*UAS-EcR-DN* ratio: 2.6 ± 0.7, n = 4 from four different brains; wildtype ratio: 1.7 ± 0.3, n = 27 from the same four brains as *UAS-EcR-DN*) (*Figure 6G–J*, *Figure 6—figure supplement 1D–D'*). Driving even stronger expression of *UAS-EcR-DN* in mushroom body neuroblasts with *inscuteable-Gal4* (referred to as *NB-Gal4*) and labeling all adult neurons with *R13F02-Gal4* (referred to as *mb2-Gal4*) (*Jenett et al., 2012*) also did not lead to the loss of α'β' neurons in the adult (*Figure 6K–N*, *Figure 6—figure supplement 1E–F*). These results indicate that EcR-DN only blocks α'β' specification when expressed in newborn mushroom body neurons, not in neuroblasts.

Given these results, we next asked whether expressing *UAS-EcR-DN* affects Mamo during development, which labels newborn postmitotic α'β' neurons (*Liu et al., 2019*); Mamo expression was lost in *UAS-EcR-DN* expressing clones driven by *mb-Gal4* (*Figure 6P*) but it was not affected by the expression of *UAS-EcR-RNAi* (*Figure 6Q*), although the RNAi was effective since we could not detect EcR protein in mushroom body neurons (*Figure 6O*, *Figure 6Q*). Given these contradictory results, we compared Mamo and EcR expression at L3. However, Mamo and EcR were mutually exclusive as EcR was not expressed in newborn α'β' neurons (see *Figure 6O''*), which precludes the possibility that EcR-DN inhibits EcR function in these neurons and might therefore act through off-target inhibition of Mamo. These results confirm the lack of EcR protein in mushroom body neuroblasts and young neurons at wandering L3 (*Figure 6—figure supplement 1G–H*), although EcR was clearly expressed in mature (mostly γ) neurons at this stage (*Figure 6—figure supplement 1H–H''*; *Lee et al., 2000*; *Liu et al., 2015*; *Marchetti and Tavosanis, 2017*). Finally, we were unable to

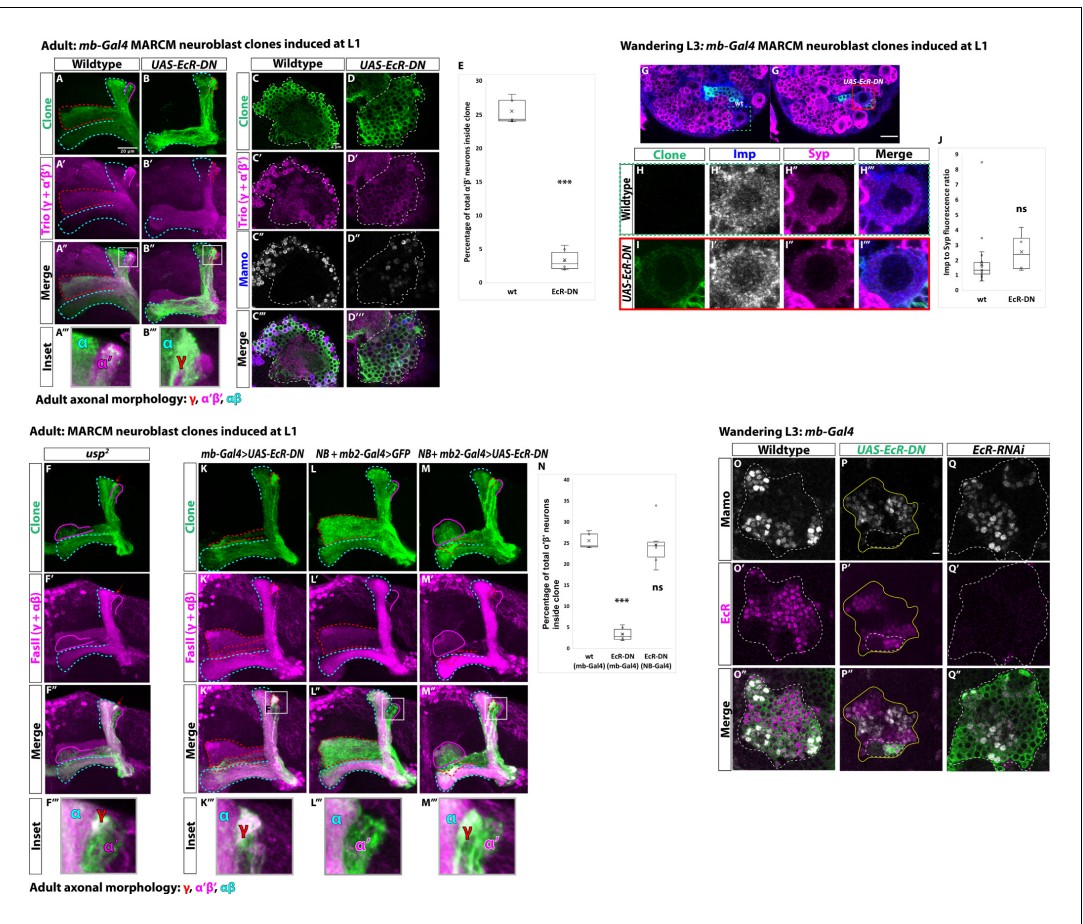

**Figure 6.** Ecdysone signaling is not necessary for α′β′ specification. (**A-B**) Representative max projections showing adult axons of clonally related neurons born from L1 stage in wildtype and *UAS-EcR-DN* conditions. *UAS-CD8::GFP* is driven by *mb-Gal4* (*OK107-Gal4*). Outlines mark GFP⁺ axons, where γ axons are outlined in red, α′β′ axons are outlined in magenta, and αβ axons are outlined in cyan. A white box outlines the Inset panel. Trio (magenta) is used to label all γ and α′β′ axons for comparison to GFP⁺ axons. (**A**) In wildtype, GFP⁺ axons are visible in all mushroom body lobes. (**B**) α′β′ axons are lost, and γ neurons do not remodel, in *UAS-EcR-DN* expressing clones. (**C-D**) Representative, single z-slices from the adult cell body region of clones induced at L1 in wildtype and *UAS-EcR-DN* conditions. *UAS-CD8::GFP* is driven by *mb-Gal4*. (**C**) Wildtype clones show the presence of strongly expressing Trio (magenta) and Mamo (blue, gray in single channel) neurons, indicative of α′β′ identity. (**D**) In *UAS-EcR-DN* clones, strong Trio and Mamo cells are not present. (**E**) Quantification of MARCM clones marked by *mb-Gal4*, which labels all mushroom body neuronal types. The number of α′β′ neurons are quantified in wildtype (n = 7, replotted from data in *Figure 1H*) and *UAS-EcR-DN* (n = 6) conditions. Plotted is the percentage of strong Mamo⁺ and GFP⁺ cells (clonal cells) versus all Mamo⁺ cells (clonal and non-clonal cells) within a single mushroom body. In wildtype, 25.5 ± 0.7% of the total strong Mamo expressing cells (α′β′ neurons) are within clones. In *UAS-EcR-DN* clones, only 3.4 ± 0.6% of α′β′ neurons are within clones. (**F**) *usp* mutant clones contain α′β′ neurons. FasII (magenta) is used to label γ and αβ lobes. Red arrow indicates unpruned γ neurons. (**G**) Representative image of an *UAS-EcR-DN* expressing neuroblast marked by *UAS-CD8::GFP* driven by *mb-Gal4* (red box) ventral to a wildtype neuroblast (green-dashed box) from the same wandering L3 stage brain, immunostained for Imp (blue, gray in single channel) and Syp (magenta). (**H**) Close-up view of wildtype neuroblast (green-dashed box in G). (**I**) Close-up view of *UAS-EcR-DN* neuroblast (red box in G). (**J**) Quantification of the Imp to Syp ratio in *UAS-EcR-DN* neuroblasts (n = 4 from four different brains) compared to wildtype (n = 27 from the same four brains as *UAS-EcR-DN* neuroblasts). (**K**) A representative adult mushroom body clone (green) induced at L1 expressing *UAS-EcR-DN* driven by *mb-Gal4*. α′β′ neurons (GFP⁺ (green), FasII- (magenta)) are not observed and γ neurons do not remodel (GFP⁺, FasII⁺, red outline). (**L**) A representative adult wildtype clone induced at L1 driven by *NB + mb2-Gal4*. All three neuron types are present, including α′β′ neurons (GFP⁺, FasII⁻, magenta outline). (**M**) α′β′ neurons are also present when *UAS-EcR-DN* is driven by *NB + mb2-Gal4* although γ neurons do not remodel. (**N**) Quantification of MARCM clones in which *UAS-EcR-DN* is driven by *mb-Gal4* (n = 6, replotted from data in E) or *NB + mb2-Gal4* (n = 6) compared to wildtype (n = 7, replotted from data in *Figure 1H*). In *UAS-EcR-DN* clones driven by *NB + mb2-Gal4*, 24.6 ± 2.1% of α′β′ neurons are within a clone, similar to wildtype. O. At L3, Mamo (gray) is expressed in young mushroom body neurons (α′β′) while EcR (magenta) can only be detected in more mature neurons (mainly γ at this stage). Note that there is no overlap between Mamo and EcR. (**P**) Expressing *UAS-EcR-DN* with *mb-Gal4* (green, white outline) leads to the loss of Mamo expression (gray) inside the clone but not in surrounding wildtype mushroom body neurons. (**Q**) In contrast, expressing *UAS-EcR-RNAi* drivenE by *mb-Gal4* abolishes EcR expression but does not affect Mamo. For E and J a two-sample, two-tailed t-test was performed. For N, a Tukey test was performed. ***p<0.001, ns: not significant. Scale bars: A, 20 μm; G, 10 μm; P, 5 μm.

*Figure 6 continued on next page*

*Figure 6 continued*

The online version of this article includes the following source data and figure supplement(s) for figure 6:

**Source data 1.** Neuron number counts for data presented in *Figure 6* and *Figure 6—figure supplement 1*.
**Source data 2.** Imp and Syp fluorescence quantification when expressing *UAS-EcR-DN*.
**Figure supplement 1.** Ecdysone signaling is not necessary for α'β' specification.

rescue the loss of α'β' neurons in *babo* mutant clones by expressing *UAS-EcR-B1* (*Figure 6—figure supplement 1I–J*), consistent with the notion that EcR does not function in α'β' specification downstream of activin signaling (see Discussion). In summary, α'β' neurons were only lost in adult clones when expressing *UAS-EcR-DN* (with *mb-Gal4*) and not in *usp* mutant clones, and α'β' neurons were still present when expressing *EcR-RNAi* in L3. Most importantly, EcR protein was not detected in Mamo[+] cells during development, although expressing *UAS-EcR-DN* blocked Mamo expression at L3. Thus, it is unlikely that ecdysone signaling is involved in α'β' specification although it might still be used later during α'β' differentiation (*Marchetti and Tavosanis, 2017*). We conclude that the loss of α'β' neurons when expressing *UAS-EcR-DN* is caused by artifactual inhibition of Mamo (see Discussion).

## Discussion

### Establishment of mushroom body neuronal identities

Mushroom body neurogenesis is unique and programmed to generate many copies of a few neuronal types. During the early stages of mushroom body development, high Imp levels in mushroom body neuroblasts are inherited by newborn neurons and translated into high Chinmo levels to specify γ identity. As in other central brain neuroblasts, as development proceeds, inhibitory interactions

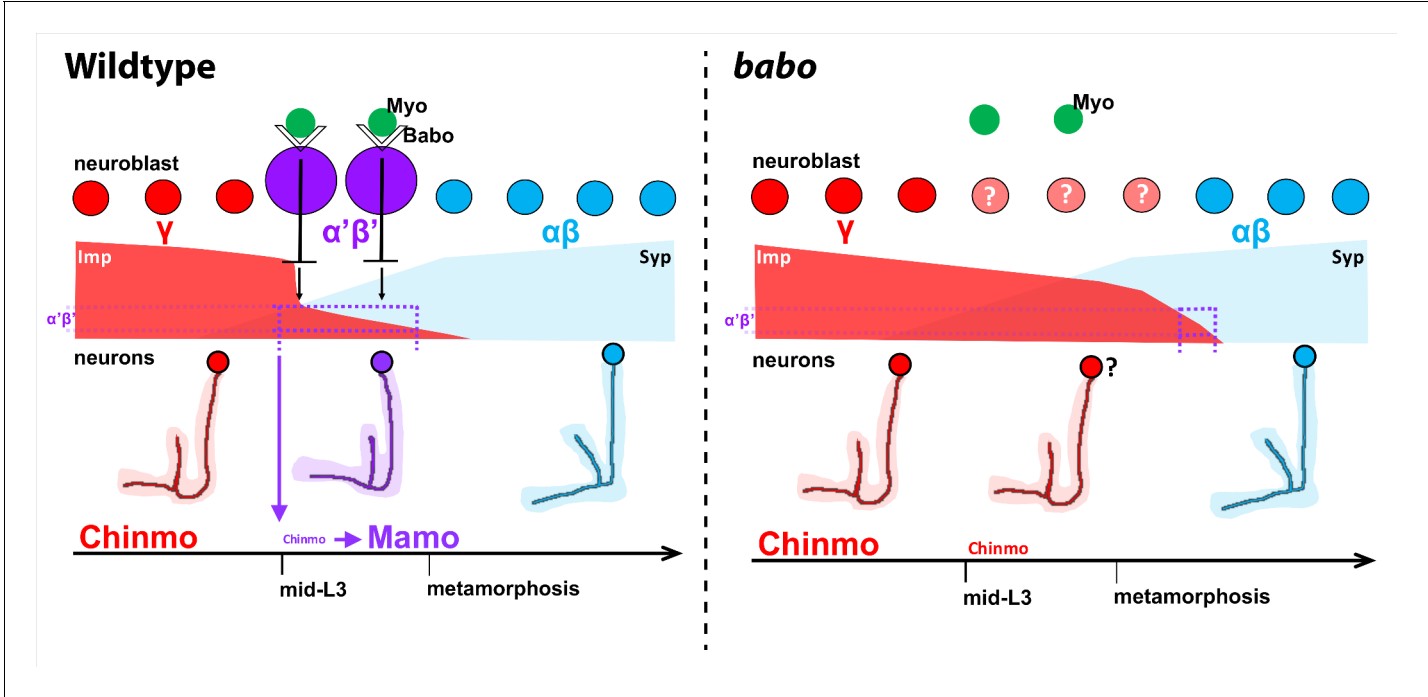

**Figure 7.** Model of how activin signaling defines the α'β' temporal identity window. In wildtype, as development proceeds, mushroom body neuroblasts incorporate an activin signal (Myo) from glia through Babo to lower the level of the intrinsic temporal factor Imp (magenta dashed line). The lower Imp levels inherited by newborn neurons leads to lower Chinmo levels to control the expression of the α'β' effector Mamo, defining the mid-temporal window (magenta dashed lines). In *babo* mutants, Imp remains higher for longer, leading to the loss of Mamo (and likely many other targets) during mid-late L3 in neurons. In this model, γ neuron numbers increase, α'β' neurons are lost, and fewer αβ neurons are produced. Nonetheless, the Imp to Syp transition still occurs, allowing for young (γ) and old (αβ) fates to be produced.

between Imp and Syp help create a slow decrease of Imp and a corresponding increase of Syp. However, at the end of the γ temporal window (mid-L3), activin signaling from glia acts to rapidly reduce Imp levels in mushroom body neuroblasts without significantly affecting Syp, establishing a period of low Imp (and thus low Chinmo in neurons) and also low Syp. This is required for activating effector genes in prospective α′β′ neurons, including Mamo, whose translation is promoted by Syp (*Liu et al., 2019*). The production of αβ identity begins when Imp is further decreased and Syp levels are high during pupation (modeled in *Figure 7*). Low Chinmo in αβ neurons is also partly regulated by ecdysone signaling through the activation of *Let-7-C*, which targets *chinmo* for degradation (*Kucherenko et al., 2012*; *Wu et al., 2012*). Based on our model, α′β′ neurons could not be rescued by knocking-down Imp in *babo* clones (*Figure 5—figure supplement 1I,N,P*), since low Imp is required for α′β′ specification while the knockdown reduces its level below this requirement. We might expect to rescue α′β′ neurons if Imp levels were specifically reduced to the appropriate levels at L3. However, reducing Imp levels might not be the only function of activin signaling, which may explain why α′β′ neurons are not simply made earlier (e.g., during L1-L2) when Imp is knocked-down.

In *babo* mutant clones, we speculate that additional γ neurons are produced at the expense of α′β′ neurons since Imp levels in neuroblasts (as well as Chinmo in neurons) are higher for a longer time during development; There was also a significant decrease in the total number of αβ neurons in *babo* mutant clones that contrasts with a recent report by Marchetti and Tavosanis that instead concluded that additional pioneer-αβ neurons are produced (*Marchetti and Tavosanis, 2019*). We believe that there is both an increase in the number of γ neurons and of the pioneer-αβ neuron subclass because pioneer-αβ neurons are the first of the αβ class to be specified (when Imp is still present at very low levels) during pupation. We speculate that pioneer-αβ neurons are produced during the extended low Imp window that we detect during pupation in *babo* clones. However, this does not leave the time for the remaining population of αβ neurons to be formed, which explains why their number is reduced.

In this study, we have focused on the three main classes of mushroom body neurons although at least seven subtypes exist: 2 γ, 2 α′β′ and 3 αβ (*Aso et al., 2014*; *Shih et al., 2019*). The subtypes are specified sequentially (*Aso et al., 2014*) suggesting that each of the three broad mushroom body temporal windows can be subdivided further, either by fine-scale reading of the changing Imp and Syp gradients, by additional extrinsic cues, or perhaps by a tTF series as in other neuroblasts.

## Temporal patterning of *Drosophila* central brain neuroblasts

Postembryonic central brain neuroblasts are long-lived and divide on average ~50 times. Unlike in other regions of the developing *Drosophila* brain, rapidly progressing series of tTFs have not yet been described in these neuroblasts (*Doe, 2017*; *Holguera and Desplan, 2018*; *Kohwi and Doe, 2013*; *Rossi et al., 2017*). Instead, they express Imp and Syp in opposing temporal gradients (*Liu et al., 2015*; *Ren et al., 2017*; *Syed et al., 2017a*). Conceptually, how Imp and Syp gradients translate into different neuronal identities through time has been compared to how morphogen gradients pattern tissues in space (*Liu et al., 2019*; *Liu et al., 2015*). During patterning of the anterior-posterior axis of the *Drosophila* embryo, the anterior gradient of the Bicoid morphogen and the posterior Nanos gradient are converted into discrete spatial domains that define cell fates (*Briscoe and Small, 2015*; *Liu et al., 2019*). Since gradients contain unlimited information, differences in Imp and Syp levels through time could translate into different neuronal types. Another intriguing possibility is that tTF series could act downstream of Imp and Syp, similarly to how the gap genes in the *Drosophila* embryo act downstream of the anterior-posterior morphogens. We have shown that another possibility is that temporal extrinsic cues can be incorporated by individual progenitors to increase neuronal diversity. In mushroom body neuroblasts activin signaling acts directly on the intrinsic program, effectively converting two broad temporal windows into three to help define an additional neuronal type. We propose that subdividing the broad Imp and Syp temporal windows by extrinsic cues may be a simple way to increase neuronal diversity in other central brain neuroblasts.

We have also shown that activin signaling times the Imp to Syp transition for mushroom body neuroblasts, similar to the function of ecdysone for other central brain neuroblasts (*Syed et al., 2017a*). In both cases however, the switch still occurs, indicating that a separate independent clock continues to tick. This role for extrinsic cues during *Drosophila* neurogenesis is reminiscent of their roles on individual vertebrate progenitors. For example, hindbrain neural stem cells progressively

produce motor neurons followed by serotonergic neurons before switching to producing glia (*Chleilat et al., 2018*; *Dias et al., 2014*). The motor neuron to serotonergic neuron switch is fine-tuned by TGFβ signaling. It would be interesting to determine if hindbrain neuronal subtypes are lost in TGFβ mutants, similar to how α′β′ identity is lost in the mushroom bodies in *babo* mutants.

## Ecdysone signaling is not necessary for α′β′ specification

The specification of α′β′ neurons begins at mid-L3 with the onset of Mamo expression (*Liu et al., 2019*). In contrast, high levels of EcR are detected in mature mushroom body neurons starting at late L3 (*Lee et al., 2000*). At this stage, both γ and α′β′ neurons already exist and new α′β′ neurons are still being generated. Thus, Mamo expression precedes EcR expression. These non-overlapping expression patterns suggest that ecdysone signaling does not regulate Mamo and therefore cannot control the specification of α′β′ neurons. Furthermore, expression of *UAS-EcR-RNAi* or mutants for *usp* do not lead to the loss of α′β′ neurons. We note that our *usp* results contradict the loss of α′β′ neuron reported by *Marchetti and Tavosanis, 2017* in *usp* clones. However, we could see α′β′ neurons in these clones based on the morphology of these neurons but the remodeling defect of γ neurons makes α′β′ neurons difficult to identify. Nevertheless, ecdysone might still function later during α′β′ differentiation, particularly during pupation when all mushroom body neurons express EcR.

We and Marchetti and Tavosanis have both shown that expression of *UAS-EcR-DN* leads to the loss of α′β′ neurons by acting in mushroom body neurons but not in neuroblasts (*Marchetti and Tavosanis, 2017*). However, EcR must be first be expressed in the target cells of interest in order to make any conclusions about ecdysone function using *UAS-EcR-DN*. Since we cannot detect EcR protein in Mamo$^+$ cells at L3, but expressing *UAS-EcR-DN* inhibits Mamo in those cells, we conclude that EcR-DN artifactually represses Mamo and leads to the loss of α′β′ neurons. This explains why expressing *UAS-EcR-B1* does not rescue α′β′ neurons in *babo* clones. However, Marchetti and Tavosanis did rescue *babo-RNAi* by expressing EcR (*Marchetti and Tavosanis, 2019*). This is likely because our experiments were performed using *babo* MARCM clones in which the loss of α′β′ neurons is much more severe than with *babo*-RNAi used in their experiments (*Figure 1—figure supplement 1Q*; *Marchetti and Tavosanis, 2019*). Indeed, when we attempted to eliminate α′β′ neurons using a validated *UAS-babo-RNAi* construct (*Awasaki et al., 2011*), γ neurons did not remodel but there was only a minor (but significant) decrease in the number of α′β′ neurons. This indicates that knocking-down *babo* with *mb-Gal4* that is only weakly expressed in neuroblasts and newborn neurons is not strong enough to inhibit α′β′ specification. Thus, we speculate that the LexA line used by Marchetti and Tavosanis (*GMR26E01-LexA*) may not be a reliable reporter for α′β′ neurons upon *babo* knockdown, and that it might be ecdysone sensitive later in α′β′ differentiation. Since EcR expression in all mushroom body neurons at L3 may be dependent on activin signaling directly in neurons, as it is in γ neurons for remodeling (*Zheng et al., 2003*), expressing *UAS-EcR-B1* together with *UAS-babo-RNAi* using *OK107-Gal4* might both reduce the effectiveness of the RNAi while also allowing for the re-expression of *GMR26E01-LexA*.

Glia are a source of the activin ligand *myo*, which is temporally expressed in brain glia starting at L3 to initiate the remodeling of mushroom body γ neurons (*Awasaki et al., 2011*) and α′β′ specification (*Figure 4*; *Marchetti and Tavosanis, 2019*). However, knocking-down Myo from glia is not as severe as removing Babo from mushroom body neuroblasts. This might be due to incomplete knockdown of *myo* or to other sources of Myo, potentially from neurons. For example, in the vertebrate cortex, old neurons signal back to young neurons to control their numbers (*Parthasarathy et al., 2014*; *Seuntjens et al., 2009*; *Toma et al., 2014*; *Wang et al., 2016*). It is also possible the Babo is activated by other activin ligands, including Activin and Dawdle (*Upadhyay et al., 2017*). An intriguing hypothesis is that the temporal expression of *myo* in glia beginning at mid-L3 is induced by the attainment of critical weight and rising ecdysone levels. It would be interesting to determine whether blocking ecdysone signaling in glia leads to the loss of α′β′ specification, similar to how blocking ecdysone reception in astrocytes prevents γ neuron remodeling (*Hakim et al., 2014*).

## Conserved mechanisms of temporal patterning

It is well established that extrinsic cues play important roles during vertebrate neurogenesis, either by regulating temporal competence of neural stem cells or by controlling the timing of temporal identity transitions (reviewed in *Kawaguchi, 2019*). Competence changes mediated by extrinsic

cues were demonstrated in classic heterochronic transplantation studies that showed that young donor progenitors produce old neuronal types when placed in older host brains (*Desai and McConnell, 2000*; *Frantz and McConnell, 1996*; *McConnell, 1988*). Recent studies show that the reverse is also true when old progenitors are placed in a young environment (*Oberst et al., 2019*).

Mechanisms of intrinsic temporal patterning are also conserved (*Alsiö et al., 2013*; *Elliott et al., 2008*; *Holguera and Desplan, 2018*; *Konstantinides et al., 2015*; *Mattar et al., 2015*; *Shen et al., 2006*). For example, vertebrate retinal progenitor cells use an intrinsic tTF cascade to bias young, middle, and old retinal fates (*Elliott et al., 2008*; *Liu et al., 2020*; *Mattar et al., 2015*). Two of the factors (Ikaros and Casz1) used for intrinsic temporal patterning are orthologs to the *Drosophila* tTFs Hb and Cas. tTF series might also exist in cortical radial glia progenitors and even in the spinal cord (*Delile et al., 2019*; *Gao et al., 2014*; *Llorca et al., 2019*; *Telley et al., 2016*; *Telley et al., 2019*). Recent results also show the importance of post-transcriptional regulation in defining either young or old cortical fates (*Shu et al., 2019*; *Zahr et al., 2018*), which can be compared to the use of post-transcriptional regulators that are a hallmark of neuronal temporal patterning in *Drosophila* central brain neuroblasts. These studies highlight that the mechanisms driving the diversification of neuronal types are conserved.

# Materials and methods

## *Drosophila* strains and MARCM

Flies were kept on standard cornmeal medium at 25°C. For MARCM experiments, embryos were collected every 12 hr. After 24 hr, L1 larvae were placed at 37°C for 2 hr for neuroblast clones or 15 min for GMC clones. To target GMCs at L3, larvae were aged for 84 hr and then placed at 37°C for 15 min. Brains were dissected from 1 to 5 day old adults.

We used the following transgenic and mutant flies in combination or recombined in this study. {} enclose individual genotypes, separated by commas. Stock numbers refers to BDSC unless otherwise stated:

{*y, w, UAS-mCD8::GFP, hsFlp; FRTG13, tub-Gal80/CYO;; OK107-Gal4* (gift from Oren Schuldiner)}, {*hsFLP, y, w; FRTG13, UAS-mCD8::GFP* (#5131)}, {*hsFLP, tubP-GAL80, w, FRT19A; UAS-mCD8::GFP/CyO; OK107-Gal4* (#44407)}, {*hsFLP, y, w, UAS-mCD8::GFP; FRT82B, tubP-GAL80/TM3, Sb$^1$; OK107-Gal4* (#44408)}, {*hsFLP, y$^1$, w\*, UAS-mCD8::GFP; tubP-GAL80, FRT40A; OK107-Gal4* (#44406)}, {*UAS-EcR.B1-DeltaC655.W650A* (#6872)}, {*y, w, FRT19A* (#1744)}, {*FRTG13, babo$^{52}$* (gift from Dr. Michael B. O'Connor)}, {*w; FRTG13* (#1956)}, {*w1118; repo-Gal4/TM3, Sb$^1$* (#7415)}, {*w; GMR71G10-GAL4* (#39604)}, {*w; GMR41C07-GAL4/TM3, Sb$^1$* (#48145)}, {*w; GMR13F02-GAL4* (#48571)}, {*w; GMR44E04-GAL4* (#50210)}, {*w\*; insc-Gal4$^{Mz1407}$* (#8751)}, {*usp$^2$/FM7a* (#31414)}, {*Met$^{27}$, gce$^{2.5K}$/FM7c, 2xTb$^1$-RFP, sn$^+$* (gift from Dr. Lynn Riddiford)}, {*y$^{d2}$, w$^{1118}$, ey-FLP; tai$^{EY11718}$ FRT40A/CyO, y$^+$* (DGRC #114680)}, {*dpy$^{ov1}$, tai$^{61G1}$, FRT40A/CyO* (#6379)}, {*w\*; smo$^{119B6}$, al$^1$, dpy$^{ov1}$, b$^1$, FRT40A/CyO* (#24772)}, {*FRT82B, svp$^1$/TM3* (gift from Tzumin Lee)}, {*y$^1$, w\*, UAS-mCD8::GFP, Smox$^{MB388}$, FRT19A/FM7c* (#44384)}, {*w\*;; UAS-p35* (#5073)}, {*y$^1$, w; Mi{PT-GFSTF.1}EcR[Ml05320-GFSTF.1]/SM6a,* (#59823)}, {*y$^1$, w\*; Pin$^{Yt}$/CyO; UAS-mCD8::GFP* (#5130)}, {*w\*;; UAS-EcR.B1* (#6469)}, {*y, w;; UAS-babo-a/TM6* (gift from Dr. Michael O'Connor)}, {*UAS-Imp-RNAi* (#34977)}, {*UAS-Imp-RM-Flag* (gift from Dr. Tzumin Lee)}, {*UAS-Syp-RNAi* (VDRC 33012, gift from Dr. Tzumin Lee)}, {*UAS-Syp-RB-HA* (gift from Dr. Tzumin Lee)}, {*y$^1$, v$^1$; UAS-myoglianin-RNAi* (#31200)}, {*w\*; OK107-Gal4/In$^4$, ci$^D$* (#854)}, {*w, UAS-EcR-RNAi* (#9326)}, {*w, UAS-EcR-RNAi* (#9327)}, {*yw, UAS-babo.Q302D* (#64293)}; {*UAS-babo-RNAi* (#44400)}.

## Immunohistochemistry and microscopy

Fly brains were dissected in ice-cold PBS and fixed for 15–20 min in 4% Formaldehyde (v/w) in 1XPBS. Following a 2 hr wash in PBST (1XPBS + 0.3% Triton X-100), brains were incubated for 1–2 days in primary antibodies diluted in PBST, followed by overnight with secondary antibodies diluted in PBST. After washes, brains were mounted in Slowfade (Life Technologies) and imaged on either a Leica SP5 or SP8 confocal. Images were processed in Fiji and Adobe Illustrator (CC18).

We used the following antibodies in this study: sheep anti-GFP (1:500, Bio-Rad #4745–1051; RRID:AB_619712), mouse anti-Trio (1:50, DSHB #9.4A anti-Trio; RRID:AB_528494), guinea pig anti-Mamo (1:200, this study, Genscript), mouse anti-FasII (1:50, DSHB #1D4 anti-Fasciclin II; RRID:AB_

528235), rat anti-Imp (1:200, this study, Genscript), rabbit anti-Syp (1:200, this study, Genscript), guinea pig anti-Dpn (1:1000, Genscript), rabbit anti-FasII (1:50, this study, Genscript), mouse anti-EcR-B1 (1:20, DSHB #AD4.4(EcR-B1); RRID:AB_528215), mouse anti-Dac2-3 (1:20, DSHB #mAbdac2-3; RRID:AB_528190), guinea pig anti-Chinmo (1:200, this study, Genscript), rat anti-Chinmo (1:200, gift from Dr. Cedric Maurange),  rat anti-DNcad (1:20, DSHB #DN-Ex #8; RRID:AB_528121), donkey anti-sheep Alexa 488 (1:500, Jackson ImmunoResearch #713-545-147; RRID:AB_2340745),  donkey anti-mouse Alexa 555 (1:400, Thermo Scientific  #A-31570; RRID:AB_2536180), donkey anti-rabbit Alexa 555 (1:400, Thermo Scientific #A-31572; RRID:AB_162543), donkey anti-rat Alexa 647 (1:400, Jackson Immunochemicals #712-605-153; RRID:AB_2340694),  donkey anti-guinea pig Alexa 647 (1:400, Jackson Immunochemicals  #706-605-148; RRID:AB_2340476),  donkey anti-rabbit 405 (1:100, Jackson Immunochemicals  #711-475-152; RRID:AB_2340616),  donkey anti-rat Cy3 (1:400, Jackson Immunochemicals  #712-165-153; RRID:AB_2340667), donkey anti-mouse 405 (1:100, Jackson Immunochemicals  #715-475-150; RRID:AB_2340839).

 Polyclonal antibodies were generated by Genscript (https://www.genscript.com/). The epitopes used for each immunization are listed below.

Mamo: amino acids 467–636 of the full length protein:
MDDRLEQDVDEEDLDDDVVVVGPATAMARGIAQRLAHQNLQRLHHTHHHAQHQHSQHHHPH SQHHHTPHHQQHHTHSDDEDAMPVIAKSEILDDDYDDEMDLEDDDEADNSSNDLGLNMKMG SGGAGGGGGVDLSTGSTLIPSPLITLPSSSAAAAAAAAAAMESQRSTPHHHHHH.

Imp: amino acids 76–455 (of isoform PB) of the full length protein:
ADFPLRILVQSEMVGAIIGRQGSTIRTITQQSRARVDVHRKENVGSLEKSITIYGNPENCTNACKRILE VMQQEAISTNKGEICLKILAHNNLIGRIIGKSGNTIKRIMQDTDTKITVSSINDINSFNLERIITVKGLIE NMSRAENQISTKLRQSYENDLQAMAPQSLMFPGLHPMAMMSTPGNGMVFNTSMPFPSCQSFAMSK TPASVVPPVFPNDLQETTYLYIPNNAVGAIIGTRGSHIRSIMRFSNASLKIAPLDADKPLDQQTERKVTIVG TPEGQWKAQYMIFEKMREEGFMCGTDDVRLTVELLVASSQVGRIIGKGGQNVRELQRVTGSVIKLPEHA LAPPSGGDEETPVHIIGLFYSVQSAQRRIRAMML.

Syp: amino acids 35-231(of isoform PA) of the full length protein:
MAEGNGELLDDINQKADDRGDGERTEDYPKLLEYGLDKKVAGKLDEIYKTGKLAHAELDERALDA LKEFPVDGALNVLGQFLESNLEHVSNKSAYLCGVMKTYRQKSRASQQGVAAPATVKGPDEDKIKKILERTG YTLDVTTGQRKYGGPPPHWEGNVPGNGCEVFCGKIPKDMYEDELIPLFENCGIIWDLRLMM.

FasII: amino acids 770–873 (of isoform PA) of the full length protein:
MHHHHHHDLLCCITVHMGVMATMCRKAKRSPSEIDDEAKLGSGQLVKEPPPSPLPLPPPVKLGG SPMSTPLDEKEPLRTPTGSIKQNSTIEFDGRFVHSRSGEIIGKNSAV.

Chinmo: amino acids 494–604 (of isoform PF) of the full length protein:
MLNVWNATKMNNKNSVNTADGKKLKCLYCDRLYGYETNLRAHIRQRHQGIRVPCPFCERTF TRNNTVRRHIAREHKQEIGLAAGATIAPAHLAAAAAASAAATAAAS NHSPHHHHHH.

## Cell counts quantification

All confocal images were taken with a step size of three microns. Using Fiji, each image was cropped to limit the area to a region containing mostly mushroom body cell bodies. In all cases, GFP$^+$ cells were manually counted. To count α′β′ neurons, images were split into their individual channels and the channel containing Mamo staining was automatically binarized to account for weak and strong Mamo expression using either Default or RenyiEntropy thresholding. Binarized images were processed further using the Watershed method to differentiate between contacting cells. The number of particles (*i.e.,* strong Mamo cells) measuring between 50-infinity squared pixels were automatically counted using the Analyze Particles function and a separate channel containing bare outlines of the counts was produced and inverted. This method automatically produced the total number of strong Mamo$^+$ cells. Individual channels were then remerged. Outlines drawn from the Analyze Particles function that overlapped with GFP$^+$ cells were defined as α′β′ neurons within a clone. In the eight cases where two mushroom body neuroblasts were labeled in a single hemisphere (wildtype:1; *babo, UAS-EcR*: 3; *babo, UAS-Syp*: 2), the total number of α′β′ neurons within clones was divided by 2.

## Imp and Syp fluorescence quantification

All brains used for quantifying Imp and Syp fluorescence values in *babo* or *UAS-EcR-DN* mutants were prepared together. Additionally, all images used for quantification were imaged using the same confocal settings for each channel. Fluorescence measurements were made in Fiji. Values for Imp and Syp were measured within the same hand-drawn area encompassing the entire neuroblast from a single z-slice.

## Statistics

Statistical tests were performed in Excel or R. The exact tests used are reported in the figure legends. In all cases, whisker plots represent the minimum value (bottom whisker), first quartile (bottom of box to middle line), inclusive median (middle line), third quartile (middle line to top of box) and maximum value (top whisker). The 'x' represents the average value. Outliers are 1.5 times the distance from the first and third quartile. Reported are averages ± standard error of mean (SEM).

## Acknowledgements

We would like to thank the fly community, the Bloomington and the DGRC stock centers for flies and reagents; Nikos Konstantinides for discussion throughout the project and comments on the manuscript, and the three reviewers for constructive feedback on our paper; Tzumin Lee for sharing data prior to publication and for discussions; Tzumin Lee, Lynn Riddiford, Oren Schuldine, Cedric Maurange and Michael B O'Connor for fly stocks and antibodies; And all the Desplan lab members for their discussion and comments on the manuscript. Funding: This work was supported by grants from NIH (R01 EY017916 and R21 NS095288) and from NYSTEM (DOH01-C32604GG) to CD. AMR was partly supported by funding from NIH (T32 HD007520), and by NYU's GSAS MacCracken Program and a Dean's Dissertation Fellowship.

## Additional information

### Competing interests

Claude Desplan: Reviewing editor, *eLife*. The other author declares that no competing interests exist.

### Funding

| Funder | Grant reference number | Author |
|---|---|---|
| National Eye Institute | R01 EY017916 | Claude Desplan |
| National Institute of Neurological Disorders and Stroke | R21 NS095288 | Claude Desplan |
| National Institutes of Health | T32 HD007520 | Anthony M Rossi |
| New York University | GSAS MacCracken Program | Anthony M Rossi |
| New York State Stem Cell Science | DOH01-C32604GG | Claude Desplan |

The funders had no role in study design, data collection and interpretation, or the decision to submit the work for publication.

### Author contributions

Anthony M Rossi, Conceptualization, Resources, Data curation, Software, Formal analysis, Funding acquisition, Validation, Investigation, Visualization, Methodology, Writing - original draft, Project administration, Writing - review and editing; Claude Desplan, Conceptualization, Resources, Supervision, Funding acquisition, Project administration, Writing - review and editing

## Author ORCIDs
Anthony M Rossi ⓘD https://orcid.org/0000-0001-9345-7939
Claude Desplan ⓘD https://orcid.org/0000-0002-6914-1413

## Decision letter and Author response
Decision letter https://doi.org/10.7554/eLife.58880.sa1
Author response https://doi.org/10.7554/eLife.58880.sa2

## Additional files

### Supplementary files
- Transparent reporting form

### Data availability

All data generated or analysed during this study are included in the manuscript and supporting files.

The following previously published dataset was used:

| Author(s) | Year | Dataset title | Dataset URL | Database and Identifier |
|---|---|---|---|---|
| Sugino K, Lee T, Liu Z, Yang C | 2015 | Opposite Imp/Syp temporal gradients govern birth time-dependent neuronal fates | https://www.ncbi.nlm.nih.gov/geo/query/acc.cgi?acc=GSE71103 | NCBI Gene Expression Omnibus, GSE71103 |

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
