## [Decision Letter]

**Acceptance summary:**

The generation of neuronal diversity often requires progenitors to produce distinct neural subtypes over time. Previous work in *Drosophila* has shown that temporal transcription factors function as a progenitor intrinsic "clock" to sequentially generate different neural subtypes. Here, Rossi and Desplan show there is also a progenitor extrinsic signaling pathway that is also required to generate temporally-distinct neural subtypes. Importantly, they document cross-talk between the intrinsic and extrinsic pathways required to generate neuronal diversity. This provides a novel paradigm for considering *Drosophila* and mammalian neural cell fate specification.

**Decision letter after peer review:**

Thank you for submitting your article "Extrinsic Activin signaling cooperates with an intrinsic temporal program to increase mushroom body neuronal diversity" for consideration by *eLife*. Your article has been reviewed by three peer reviewers, including Chris Q Doe as the Reviewing Editor and Reviewer #1, and the evaluation has been overseen by K VijayRaghavan as the Senior Editor.

The reviewers have discussed the reviews with one another and the Reviewing Editor has drafted this decision to help you prepare a revised submission.

As the editors have judged that your manuscript is of interest, but as described below that additional experiments are required before it is published, we would like to draw your attention to changes in our revision policy that we have made in response to COVID-19 (https://elifesciences.org/articles/57162). First, because many researchers have temporarily lost access to the labs, we will give authors as much time as they need to submit revised manuscripts. We are also offering, if you choose, to post the manuscript to bioRxiv (if it is not already there) along with this decision letter and a formal designation that the manuscript is "in revision at *eLife*". Please let us know if you would like to pursue this option.

Summary:

This manuscript from Rossi and Desplan addresses the integration of intrinsic and extrinsic cues in the specification of temporal identity within the *Drosophila* mushroom body neuroblast (MBNB) lineages, which sequentially produce four subtypes of neurons: γ, α’/β’, p.α’/β’ and a/b. The mechanisms generating neuronal diversity are important and of general interest; the integration of progenitor intrinsic and extrinsic cues is relatively novel.

The manuscript builds on closely related work from Marchetti and Tavosanis (2017, 2019). Both labs show that the TGFb signaling pathway is required for α’/β’ adult neuron identity, but via highly divergent mechanisms. Marchetti proposes TGFb drives expression of the steroid hormone receptor EcR which is required for consolidating α’/β’ fate in adult neurons (but not for their specification); when the pathway is compromised, α’/β’ neurons switch to a later-born pioneer.α’/β’ fate. In contrast, Rossi proposes TGFb represses the intrinsic temporal factor Imp in neuroblasts, with low Imp levels specifying α’/β’ identity; when the pathway is compromised, the α’/β’ neurons switch to an earlier-born high Imp γ neuron identity.

All reviewers felt the current manuscript contained well-designed experiments, convincing figures, and that the logical flow made it easy to follow. All reviewers felt the paper was appropriate for *eLife* based on the journal's policy of reviewing papers where an overlapping paper has been recently published with the goals of (a) independently replicating results, or (b) correct previous findings. Nevertheless, all reviewers felt this manuscript failed to resolve conflicts in data and interpretation with prior papers, particularly the role of EcR and the difference between specification and consolidation. Thus, the large majority of comments below ask for additional experiments or text changes to reconcile the two sets of results.

Essential revisions:

1) The role of EcR signaling. This is central to the Marchetti model, but excluded from the Rossi model. The authors should put more effort into resolving the different conclusions on EcR signaling in specifying α’/β’ neuron identity. Some experiments give similar results (e.g. EcR-DN expression reduces α’/β’ neurons), however, the global interpretation is quite different, which makes the current work more significant. Unfortunately, the differences in interpretation are not explained or discussed, which leave the reader in a perplexed mood, because the main conclusions of the two papers do not seem compatible.

a) Marchetti et al. show that TGFb signaling induces expression of EcRB1, and that forced expression of EcRB1 can rescue a *babo* mutant phenotype; the α’/β’ neurons are restored. This is strong evidence for the role of EcRB1 in TGFb specification of α’/β’ identity. The Rossi manuscript also shows that the TGFb pathway is required for α’/β’ fate, but independent of EcR. How can these opposing results be reconciled?

b) The fact that the α'/β' marker Mamo is still expressed when EcR-RNAi is mis-expressed does not necessarily means that ecdysone signaling is not required for α’β’ specification. It is well-documented that EcRDN and EcR-RNAi mis-expression can lead to opposite phenotypes (PMID: 16354717, 25126791, 28394252). Indeed, EcR-DN prevents activation of ecdysone signaling induced by the production of ecdysone, but leaves intact the repressive ability of unbound EcR. In contrast, EcR-RNAi prevents both activation of ecdysone signaling and the repressive role of EcR in absence of ecdysone. One possibility is that Mamo, in the absence of ecdysone is repressed in neurons by unbound EcR, and that ecdysone signaling activates Mamo expression by relieving the EcR-mediated repression. If this is true, one may see precocious expression of Mamo in early L3 upon EcR-RNAi expression (similar to the UAS-babo-Act phenotype). The authors should investigate this to clarify the point.

c) Major differences are observed in the experiments removing the EcR co-factor Usp. Marchetti, 2017 showed *usp* clones lack α’/β’ neurons; here *usp* clones show normal α’/β’ neurons. Are both labs using the same reagent? Can the authors show that their *usp* mutant clones lack USP protein?

2) The primary novel conclusion in the present manuscript is that Myo signaling specifies α’/β’ neurons via regulating Imp levels, but the data to support this is rather slim. The only supportive data is the elevated Imp level in *babo* mutant neuroblast.

a) Importantly, they failed to rescue the α’/β’ neuron identity by forced expression of Imp in *babo* mutant clones. How is this consistent with their model?

b) Furthermore, the authors show that overexpression of EcR-DN leads to suppression of α’/β’ neuron specification (Figure 6E), but in the same figure, they also show that the Imp levels are not changed in the neuroblasts (Figure 6H and J), which suggests that the changing Imp levels in neuroblasts may not be required to change α’/β’ neuron specification.

c) Also, the authors demonstrate that TGFb signaling intersects with the Imp/Syp temporal patterning system by showing that Imp remains expressed at higher levels in MB NBs that are mutant for babo. The maintenance of higher levels of Imp is supposed to be sufficient to prevent expression of Mamo. Since Chinmo is regulated by the Imp/Syp module and that Mamo is activated by low levels of Chinmo, one should expect high levels of Chinmo to be maintained in late larval neurons produced in the *babo* mutant context. Is this observed? The Marchetti study says that they don't see any obvious changes.

d) If TGFb signaling is indeed required to terminate the first temporal window and activate the second temporal window then, γ neurons should still be generated in late L3 MB neuroblasts that are mutants for babo. As this is such a critical point in their model, the authors should do a precise time course to determine how long MB neuroblasts continue to generate γ neurons when mutant for *babo* (compared to wt). This should be checked using Abrupt as a marker for γ neurons, which seems to be more specific than Trio.

e) Related to the previous point: the evidence that TGFb signaling intersects with progression of the Imp/Syncrip temporal patterning system remains thin (slightly higher levels of Imp in late larval and pupal MB neuroblasts that are mutant for babo). Further evidence is needed, to confirm that TGFb signaling is responsible for creating a novel temporal window, as opposed to a role in fate consolidation as proposed by Marchetti.

f) Finally, they showed that either knockdown or overexpression of Imp using mb-GAL4 leads to loss of Mamo, a marker for α’/β’ neurons. mb-GAL4 is strongly expressed in mushroom body neurons and is weakly expressed in neuroblasts, so the manipulation of Imp level is not only done in neuroblasts but also in neurons, and thus it is possible that the Imp, which appears to also be expressed in post mitotic neurons (Figure S3D), is actually required in already specified α’/β’ neurons to maintain their identity, consistent with the Marchetti model.

3) Marchetti show that TGFb signaling acts in neuroblasts, but is not required for α’/β’ neuron specification during larval stages, nor for the normal levels of the early temporal factors Chinmo and Abrupt. Rather, TGFb signaling is required to stabilize/consolidate α’/β’ identity in adults. In this work, loss of TGFb signaling increases Imp temporal identity levels, which should alter Chinmo and Abrupt levels. Can you resolve this discrepancy?

4) The authors state in the text and show in the final model figure that loss of TGFb signaling leads to a loss of mid-born α’/β’ neurons (well supported by multiple experiments) and an expansion of early-born γ neurons, "although not significantly". Either more n's need to be added to (potentially) reveal significance, or the figure and conclusions need to be toned down. Showing a doubling of γ neurons in the model figure when they are 'not significantly' increased is a stretch.

5) Marchetti observe EcR-B1 expression in all α’β’ neurons. This manuscript shows EcR-B1 is not expressed in Mamo+ α’/β’ neurons. This is a puzzle that should be resolved. Perhaps by checking that C305-GAL4 and Mamo are expressed in the same set of mushroom body neurons in late L3?

6) An important point that remains unresolved and that has not been investigated nor discussed is what controls the temporal expression of Myo in glia after critical weight is achieved at mid-larval stage. An attractive hypothesis is that it is induced by the mid-L3 pulses of ecdysone. This can be easily investigated using available Myo-GAL4-driven GFP expression lines and looking at mid 3rd instar. Resolving this point would make the whole study more attractive, and different from the Marchetti one.

7) Marchetti shows loss of TGFb signaling transforms α’/β’ neurons to later-born pioneer α’/β’ neurons; this work shows that loss of TGFb signaling transforms α’/β’ neurons to earlier-born γ neurons. Please discuss.

---

## [Author Response]

Essential revisions:1) The role of EcR signaling. This is central to the Marchetti model, but excluded from the Rossi model. The authors should put more effort into resolving the different conclusions on EcR signaling in specifying α’/β’ neuron identity. Some experiments give similar results (e.g. EcR-DN expression reduces α’/β’ neurons), however, the global interpretation is quite different, which makes the current work more significant. Unfortunately, the differences in interpretation are not explained or discussed, which leave the reader in a perplexed mood, because the main conclusions of the two papers do not seem compatible.

To make sure that the interpretation of the differences is clear, we have rewritten and added additional text to explain the differences between our work and the work from Marchetti and Tavosanis. We exclude EcR from our model because our data suggest that only EcR-DN, but not EcR knockdown or *usp* mutants, blocks α’β’ specification. EcR-DN blocks α’β’ specification by inhibiting Mamo. Obviously, EcR must be expressed in these cells for EcR-DN to demonstrate function. However, EcR is not expressed in Mamo+ cells that are destined to become α’β’ neurons during L3. In addition, Mamo starts being expressed in presumptive α’β’ neurons from mid-L3 while EcR is only activated later at wandering L3, another point that we highlight in the text. Following this logic, we conclude that EcR-DN must artificially repress Mamo during α’β’ specification. We speculate that Mamo may be regulated by EcR in other contexts and that this regulation is hijacked by expressing *UAS-EcR-DN* with *OK107-Gal4*.

a) Marchetti et al. show that TGFb signaling induces expression of EcRB1, and that forced expression of EcRB1 can rescue a babo mutant phenotype; the α’/β’ neurons are restored. This is strong evidence for the role of EcRB1 in TGFb specification of α’/β’ identity. The Rossi manuscript also shows that the TGFb pathway is required for α’/β’ fate, but independent of EcR. How can these opposing results be reconciled?

We agree that this is a main difference between the two studies and have addressed it by adding a new result and text in our Results and Discussion sections. The most parsimonious explanation is that the differences in the results are a consequence of methods used. First, we failed to rescue *babo* mutant clones while they rescued a *babo-RNAi*. *babo* mutant clones lead to an almost complete loss of α’β’ neurons (while, in their manuscript, expressing *babo-RNAi* using *OK107-Gal4* reduces the number of α’β’ neurons by about 2/3). Second, we quantified the loss of α’β’ neurons in adult clones using two molecular markers (high Mamo and Trio) while they used a LexA driver (*GMR26E01-LexA*). We speculate that this LexA driver is ecdysone sensitive (indirectly). Indeed, all mushroom body neurons express EcR during the pupal stages. If Activin signaling activates EcR expression in all mature mushroom body neurons as it does in γ neurons (independent of its role in mushroom body neuroblasts), then this LexA driver may not be effectively activated late. By expressing *UAS-EcR-B1* and *babo-RNAi* together, they might rescue expression of the LexA driver. In support of this, we now show that by knocking-down *babo* with *OK107-Gal4,* that the majority of α’β’ neurons are present in the adult using Mamo and Trio while g neurons do not remodel in these same brains. Therefore, we speculate that *GMR26E01-LexA* is not a reliable marker for α’β’ neurons in this context.

**Author response image 1. sa2fig1:** Representative images showing the presence of α’β’ neurons based on strong Mamo (gray) and Trio (magenta) expression in adult mushroom body neurons labeled by *mb-Gal4* driving *UAS-CD8::GFP*. N-N’’’. Images highlighting the vertical mushroom body axons. α’ axons are Trio^+^ and GFP^+^. FasII (gray) labels α axons. O-O’’. The majority of α’β’ neurons are still present following expression of *UAS-babo-RNAi* based on strong Mamo and Trio expression. P-P’’’. The vertical lobes indicate that γ neurons do not remodel (FasII^+^/Trio^+^/GFP^+^) and that α’β’neurons are still present (α’ lobe that is FasII^-^/Trio^+^/GFP^+^) following *UAS-babo-RNAi* expression. Q. Quantification of the number of α’β’ neurons following expression of *UAS-babo-RNAi* (n=6) compared to wildtype controls (n=6). A two-sample, two-tailed t-test was performed.

b) The fact that the α’/β’ marker Mamo is still expressed when EcR-RNAi is mis-expressed does not necessarily means that ecdysone signaling is not required for α’β’ specification. It is well-documented that EcRDN and EcR-RNAi mis-expression can lead to opposite phenotypes (PMID: 16354717, 25126791, 28394252). Indeed, EcR-DN prevents activation of ecdysone signaling induced by the production of ecdysone, but leaves intact the repressive ability of unbound EcR. In contrast, EcR-RNAi prevents both activation of ecdysone signaling and the repressive role of EcR in absence of ecdysone. One possibility is that Mamo, in the absence of ecdysone is repressed in neurons by unbound EcR, and that ecdysone signaling activates Mamo expression by relieving the EcR-mediated repression. If this is true, one may see precocious expression of Mamo in early L3 upon EcR-RNAi expression (similar to the UAS-babo-Act phenotype). The authors should investigate this to clarify the point.

We are aware that EcR-DN and EcR-RNAi have different modes of action. Although the suggestion by the reviewer is logical, the underlying assumption is that EcR is normally expressed in Mamo+ cells. We have shown that this is not the case, as EcR and Mamo do not overlap. If the above model was correct (i.e., unbound EcR inhibits Mamo), then EcR must be expressed in Mamo+ cells, which it is not. Furthermore, this model would require that EcR be expressed before Mamo in mushroom body neurons during development and again, this is not the case since Mamo is expressed in presumptive α’β’ neurons from mid-L3 while EcR can only be detected later at wandering L3. We have added text to our Results and Discussion section highlighting this point.

c) Major differences are observed in the experiments removing the EcR co-factor Usp. Marchetti, 2017 showed usp clones lack α’/β’ neurons; here usp clones show normal α’/β’ neurons. Are both labs using the same reagent? Can the authors show that their usp mutant clones lack USP protein?

We initially reported clones of the *usp2* (amorphic) allele while they used (hypomorphic) *usp3* allele. We therefore made *usp3* mutant clones, which we now report also contain α’β’ neurons (and unpruned γ neurons) (See Figure 6—figure supplement 1C-C’’’). The difference in interpretation may be due to the fact that in the *usp* clones, γ neurons do not remodel, which can make identifying α’β’ neurons based on morphology alone quite difficult. Nonetheless, from the single image published in Marchetti and Tavosanis’ paper, it is clear that their *usp* mutant clones induced at L1 contain α’β’ neurons marked by GFP+/FasII- are present. The reviewer asked us to show that our *usp* mutant clones lack Usp protein: all of our *usp* neuroblasts clones induced at L1 contain unpruned γ neurons (see Figure 6F, Figure 6—figure supplement 1C-C’’’), demonstrating that *usp* is indeed missing. We have added text to our Discussion describing our interpretation of the Marchetti and Tavosanis results. We have communicated with these authors multiple times to explain our interpretation.

2) The primary novel conclusion in the present manuscript is that Myo signaling specifies α’/β’ neurons via regulating Imp levels, but the data to support this is rather slim. The only supportive data is the elevated Imp level in babo mutant neuroblast.

We agree that the conclusion that Activin signaling interacts with the intrinsic Imp and Syp temporal gradients is the most novel concept in our paper. Our work not only confirms that Activin signaling is necessary to specify α’β’ neurons but also shows how two different temporal mechanisms interact to increase neuronal diversity.

We believe that we provide a very strong proof that Imp levels in mushroom body neuroblasts are regulated by Activin signaling: Imp levels in mutant neuroblast clones are very significantly increased and this was internally controlled with the levels of Imp in neighboring wildtype neuroblasts. Furthermore, this is further supported by the high levels of Chinmo in *babo* clones at the time α’β’ neurons should be produced (see image S3C). We feel that these are more than slim evidence.

a) Importantly, they failed to rescue the α’/β’ neuron identity by forced expression of Imp in babo mutant clones. How is this consistent with their model?

We guess the reviewers meant to comment on our rescue of *babo* clones by reducing *Imp* levels, which did not rescue α’β’ specification (Figure 5—figure supplement 1). However, we show (and it is well documented) that *Imp-RNAi* expression leads to the loss of α’β’ neurons and our results are entirely consistent with our proposed model that low imp (but not ‘no Imp’) is required for α’β’ specification. Given our model, we think that reducing the levels of *Imp* specifically at L3 in *babo* mutant clones (but still inducing clones at L1) might rescue the loss of α’β’ neurons. This would require us to perform long and technically difficult experiments that were in progress but were interrupted by the lockdown. We now include text in our Discussion addressing this experiment and our model.

b) Furthermore, the authors show that overexpression of EcR-DN leads to suppression of α’/β’ neuron specification (Figure 6E), but in the same figure, they also show that the Imp levels are not changed in the neuroblasts (Figure 6H and J), which suggests that the changing Imp levels in neuroblasts may not be required to change α’/β’ neuron specification.

As described above, we suggest that EcR-DN blocks α’β’ specification by acting late in postmitotic neurons, not in neuroblasts. We showed that expressing EcR-DN in neuroblasts (with *Insc-Gal4*) does not inhibit α’β’ specification (Figure 6M). Thus, Imp levels in neuroblasts should not be affected by expressing EcR-DN: Imp (and Syp) levels are inherited by neurons at birth, which is where these RNA-binding proteins act to regulate neuronal specification. In addition, EcR is not expressed in mushroom body neuroblasts or newborn postmitotic neurons (Mamo+), explaining why ecdysone signaling can only act in older postmitotic neurons. As explained above, EcR-DN acts by artificially inhibiting Mamo in newborn neurons. We have added text to our Results section highlighting this point.

c) Also, the authors demonstrate that TGFb signaling intersects with the Imp/Syp temporal patterning system by showing that Imp remains expressed at higher levels in MB NBs that are mutant for babo. The maintenance of higher levels of Imp is supposed to be sufficient to prevent expression of Mamo. Since Chinmo is regulated by the Imp/Syp module and that Mamo is activated by low levels of Chinmo, one should expect high levels of Chinmo to be maintained in late larval neurons produced in the babo mutant context. Is this observed? The Marchetti study says that they don't see any obvious changes.

We agree with the reviewers that our model indicates that Chinmo levels should be higher in *babo* mutant neurons at L3. We had done experiments to test this and were now able to analyze them and they are now included in Figure 2—figure supplement 1C-C’’’. The results indicate that Chinmo levels are indeed higher in mutant neurons while they are low in the adjacent wildtype neurons.

d) If TGFb signaling is indeed required to terminate the first temporal window and activate the second temporal window then, γ neurons should still be generated in late L3 MB neuroblasts that are mutants for babo. As this is such a critical point in their model, the authors should do a precise time course to determine how long MB neuroblasts continue to generate γ neurons when mutant for babo (compared to wt). This should be checked using Abrupt as a marker for γ neurons, which seems to be more specific than Trio.

We show that Activin signaling is not absolutely required to terminate the first temporal window: Imp levels still decline and Syp levels still increase through time (although at a significantly lower pace) independently of Activin signaling (Figure 2—figure supplement 1J-J’). We attempted to test whether there was an increase in the number of γ neurons as this is an important component of our model but could only demonstrate a trend upwards in the adult, likely because the timing of clone induction dramatically affects variability of the number of γ neurons, while it does not affect later time windows. Unfortunately, a γ specific marker at L3 does not exist (e.g., Abrupt labels both γ and α’β’ neurons at L3). To attempt to answer whether the γ window is maintained for longer, we now include a figure showing that Chinmo levels are higher in *babo* mutant clones compared to surrounding wildtype neurons at wandering L3 (Figure 2—figure supplement 1C-C’’’). These mutant neurons do not express Mamo but wildtype neurons do.

e) Related to the previous point: the evidence that TGFb signaling intersects with progression of the Imp/Syncrip temporal patterning system remains thin (slightly higher levels of Imp in late larval and pupal MB neuroblasts that are mutant for babo). Further evidence is needed, to confirm that TGFb signaling is responsible for creating a novel temporal window, as opposed to a role in fate consolidation as proposed by Marchetti.

In order to determine that Imp is affected by Activin signaling we made MARCM clones for *babo* and characterized the levels of Imp compared to surrounding wt neuroblasts in the same brains. This clearly demonstrated a ~1.5-fold increase in Imp levels in *babo* mutant neuroblasts, which was quantified in Figure 2D and Figure 2—figure supplement 1B ). Furthermore, our data agree with Marchetti and Tavosanis who interpret their data, like us, that Babo acts in neuroblasts and not in neurons, although they do not investigate whether Imp and/or Syp levels are altered. They do argue that the α’β’ fate requires EcR signaling in neurons, i.e the two pathways would act in different cell types (neuroblasts versus neurons). Our study provides an answer to this problem as we propose that Activin signaling acts on mushroom body neuroblasts to affect Imp levels, which are inherited by newborn neurons. We have revised our text to make this point clearer.

f) Finally, they showed that either knockdown or overexpression of Imp using mb-GAL4 leads to loss of Mamo, a marker for α’/β’ neurons. mb-GAL4 is strongly expressed in mushroom body neurons and is weakly expressed in neuroblasts, so the manipulation of Imp level is not only done in neuroblasts but also in neurons, and thus it is possible that the Imp, which appears to also be expressed in post mitotic neurons (Figure S3D), is actually required in already specified α’/β’ neurons to maintain their identity, consistent with the Marchetti model.

The reviewers are correct in that Imp is expressed in neurons. In fact, the original model from Tzumin Lee argues that Imp and Syp are expressed in gradients in neuroblasts but carry out their temporal patterning function in neurons (since they are inherited). These Imp levels in neurons control Chinmo translation and thus other downstream targets. When we express *Imp-RNA* with *OK107-Gal4* (*mb-Gal4*), we are in fact lowering *Imp* in both neuroblasts and neurons. This does not change the conclusion that low Imp levels in neuroblasts, which would normally be inherited from neuroblasts into neurons, are needed for low Chinmo levels and thus for Mamo expression in α’β’ neurons. Our new data show that Chinmo levels are elevated in neurons at L3 in *babo* mutant clones, which support our interpretation. Low Imp levels inherited from neuroblasts are what help define α’β’ neurons with everything else occurring downstream of this temporal factor. We have added text to our Discussion that better describes how Imp and Syp levels in neuroblasts are inherited by newborn neurons to control temporal identity.

3) Marchetti show that TGFb signaling acts in neuroblasts, but is not required for α’/β’ neuron specification during larval stages, nor for the normal levels of the early temporal factors Chinmo and Abrupt. Rather, TGFb signaling is required to stabilize/consolidate α’/β’ identity in adults. In this work, loss of TGFb signaling increases Imp temporal identity levels, which should alter Chinmo and Abrupt levels. Can you resolve this discrepancy?

It is our understanding that Marchetti and Tavosanis argue that Activin signaling at L3 leads to the expression of EcR in differentiating (but already specified) α’β’ neurons. They do not provide a model for how Activin signaling in neuroblasts is transmitted to neurons leading to EcR expression but the activation of EcR must occur when ecdysone is present. Although they did stain for Chinmo in *babo* mutant clones at L3, it is unclear from this single image they show whether Chinmo levels are altered. Nonetheless, we did not check Abrupt (for which we do not have a good antibody anyway) since this marker labels both γ and α’β’ neurons at L3. However, we now include data in Figure 2—figure supplement 1C’C’’’ that shows higher Chinmo levels and loss of Mamo in *babo* mutant neurons, which is consistent with our model that in the absence of Activin signaling, higher Imp levels (in neuroblasts) at L3 are inherited by neurons and translated into higher Chinmo levels, which leads to the loss of Mamo and of α’β’ specification.

4) The authors state in the text and show in the final model figure that loss of TGFb signaling leads to a loss of mid-born α’/β’ neurons (well supported by multiple experiments) and an expansion of early-born γ neurons, "although not significantly". Either more n's need to be added to (potentially) reveal significance, or the figure and conclusions need to be toned down. Showing a doubling of γ neurons in the model figure when they are 'not significantly' increased is a stretch.

We agree with the reviewers that our suggestion that there are additional γ neurons is based on indirect data (i.e., higher Chinmo levels at L3, loss of Mamo at L3, no change in the number of mushroom body neurons but a loss of α’β’ neurons and significant decrease in αβ neurons). We have corrected this in our model figure (Figure 7). We have also added the word, “likely” when we suggest that additional γ neurons are produced in the text.

5) Marchetti observe EcR-B1 expression in all α’β’ neurons. This manuscript shows EcR-B1 is not expressed in Mamo+ α’/β’ neurons. This is a puzzle that should be resolved. Perhaps by checking that C305-GAL4 and Mamo are expressed in the same set of mushroom body neurons in late L3?

We show that EcR-B1 is not expressed in newborn Mamo+ α’β’ neurons at L3. This agrees with Figure 2D of the 2017 paper from Marchetti and Tavosanis in which the majority of *c305-Gal4* neurons are EcR-B1 negative at L3. It is correct that α’β’ neurons express EcR late, during early pupal stages, but at this point all mushroom body neurons express EcR, the α’β’ window has closed, and αβ neurons are being specified. We also now discuss in the text that as α’β’ neurons mature, they first express Mamo and then later EcR, which is another reason why EcR cannot affect Mamo expression and α’β’ specification.

6) An important point that remains unresolved and that has not been investigated nor discussed is what controls the temporal expression of Myo in glia after critical weight is achieved at mid-larval stage. An attractive hypothesis is that it is induced by the mid-L3 pulses of ecdysone. This can be easily investigated using available Myo-GAL4-driven GFP expression lines and looking at mid 3rd instar. Resolving this point would make the whole study more attractive, and different from the Marchetti one.

We agree that the question of what times the expression of Myo (and potentially other Activin ligands) from glia at L3 is very interesting. We highlight in the text that it has already been shown that *myo* is temporally expressed in central brain glia starting at mid-L3 (Awasaki et al., 2011). We also now include in the text the hypothesis that attainment of critical weight (and an increase in ecdysone) are responsible for this expression pattern of *myo*.

Attempting to determine whether ecdysone activates *myo* expression in glia is not straightforward, as this would require blocking EcR without using the Gal4-UAS system in glia and then combining this with Myo-Gal4 as suggested. This would require new tools and would likely lead to new avenues of research and additional questions that would go beyond the scope of this work. We plan on addressing the interaction between extrinsic cues, glia, and neuronal specification in future work.

7) Marchetti shows loss of TGFb signaling transforms α’/β’ neurons to later-born pioneer α’/β’ neurons; this work shows that loss of TGFb signaling transforms α’/β’ neurons to earlier-born γ neurons. Please discuss.

Indeed, Marchetti and Tavosanis concluded that additional pioneer-αβ neurons were born while we clearly show that the number of αβ neurons is significantly decreased, while γ neurons may increase, in *babo* mutant clones. The two interpretations are not at odds since higher than normal Imp levels during pupation might extend the window for pioneer-αβ neuron specification (since they are the first αβ neuronal type to be specified) while the majority of remaining later-born αβ will not have time to be generated before birth. This would both decrease the overall number of αβ neurons as we suggest but could also lead to an increase in the pioneer-αβ population. We have added text to our Discussion describing these possibilities.